# Air pollution slows down surface warming over the Tibetan Plateau

Aolin Jia[1], Shunlin Liang[1], Dongdong Wang[1], Bo Jiang[2], Xiaotong Zhang[2]

[1]Department of Geographical Sciences, University of Maryland, College Park, MD, 20742, USA
[2]State Key Laboratory of Remote Sensing Science, Faculty of Geographical Science, Beijing Normal University, Beijing, 10085, China

*Correspondence to*: Shunlin Liang (sliang@umd.edu)

**Abstract.** The Tibetan Plateau (TP) plays a vital role in regional and global climate change. The TP has been undergoing significant surface warming since 1850, with an air temperature increase of 1.39 K and surface solar dimming resulting from decreased incident solar radiation. The causes and impacts of solar dimming on surface warming are unclear. In this study, long-term (from 1850–2015) surface downward radiation datasets over the TP are developed by integrating 18 Coupled Model Intercomparison Project Phase 5 (CMIP5) models and satellite products. The validation results from two ground measurement networks show that the generated downward surface radiation datasets have a higher accuracy than the mean of multiple CMIP5 and the fused datasets of reanalysis and satellite products. After analyzing the generated radiation data with four air temperature datasets, we found that downward shortwave radiation (DSR) remained stable before 1950 and then declined rapidly at a rate of -0.53 W m$^{-2}$ per decade and that the fastest decrease in DSR occurs in the southeastern TP. Evidence from site measurements, satellite observations, reanalysis, and model simulations suggested that the TP solar dimming was primarily driven by increased anthropogenic aerosols. The TP solar dimming is stronger in summer, at the same time that the increasing magnitude of the surface air temperature is the smallest. The cooling effect of solar dimming offsets surface warming on the TP by $0.80 \pm 0.28$ K ($48.6 \pm 17.3\%$) in summer since 1850. It helps us understand the role of anthropogenic aerosols in climate warming, and highlights the need for additional studies to be conducted to quantify the influence of air pollution on regional climate change over the TP.

## 1 Introduction

The Tibetan Plateau (TP), the so-called Third Pole, covers an area of approximately $2.65 \times 10^2$ km$^2$ and has an average elevation of more than 4000 m. It contains the largest ice mass outside of the polar regions (Yao et al., 2007) which supplies several major rivers that sustain billions of people in China and South Asia, dominating regional social stabilization and development. The TP is a weak heat sink in winter but a strong heat source in summer and dominates the atmospheric circulation (Wu et al., 2015). The mechanical and thermal forcing of the large-scale orography is crucial for the formation of the Asian summer monsoon (Boos and Kuang, 2010) and water and heat exchange between the Pacific Ocean and Eurasia (Wu et al., 2012). The TP anticyclone transports water vapor and chemical gases into the lower stratosphere (Fu et al., 2006).

Therefore, the local climate pattern over the TP plays a vital role in the climate in southern China (Xu et al., 2013), the boreal climate (Sampe and Xie, 2010), and global climate change (Cai et al., 2017).

The TP is currently undergoing significant climate change (Yao et al., 2018), such as increased surface air temperature (Kuang and Jiao, 2016) and downward longwave radiation (DLR), as well as decreased downward solar radiation (DSR) (You et al., 2010) which is a solar dimming phenomenon that can impact surface air temperatures (Wild et al., 2007) and precipitation

(Wild, 2009). However, the causes of solar dimming over the TP are not yet conclusive (Kuang and Jiao, 2016;Xie and Zhu, 2013;Xie et al., 2015). Changes in DSR are mainly controlled by atmospheric clouds and aerosols (Liang et al., 2010) at century-level scales. You et al. (2013) suggested that aerosols have played an important role in solar dimming over the TP in recent decades, while Tang et al. (2011) speculated that solar dimming over the TP might be caused by cloud cover changes that have a comparable influence on solar dimming to that of aerosol loading changes. Although some studies have suggested

that brown clouds that are formed due to aerosols over the Indian Ocean and Asia (Ramanathan et al., 2007) might be transported to the TP by the summer monsoon, Yang et al. (2012) and (2014) contended that the main drivers are deep convective clouds and atmospheric water vapor, and that aerosol radiative forcing is too small to result in the decreased DSR over the TP. These studies clearly show contradictory conclusions regarding the proper attribution of the solar dimming; therefore, the roles of aerosols and clouds in solar dimming still need to be determined. Moreover, some of these studies were

mainly based on ground measurements at a limited number of sites that cannot represent the entire TP.  Furthermore, the surface observed sunshine duration data that were used for estimating DSR (Wang, 2014;Yang et al., 2006) were not available over the TP until the 1960s. Regardless, the statistical model can hardly capture the low aerosols' influences on surface solar radiation by sunshine duration because sunshine duration has a lower sensitivity than DSR to atmospheric turbidity changes that is estimated by aerosol optical depth (AOD) (Manara et al., 2017). Thus, considering that remote sensing has been

developed for several decades, it provides a valuable opportunity to employ satellite observations to monitor spatiotemporal variations at regional scales.

The radiative effect of anthropogenic aerosols has not yet been well quantified over the TP, which is necessary for understanding the role of anthropogenic aerosols in climate warming, and revealing impacts of human activities in remote areas. Aerosols have a net cooling effect on the global temperature with higher uncertainty from Intergovernmental Panel on

Climate Change (IPCC) report (Stocker et al., 2013), whereas Andreae et al. (2005) have suggested that current aerosol loading may cause a hot future. Even Gettelman et al. (2015) contended that the net effect of aerosols on surface temperature can be neglected, Samset et al. (2018) pointed out that aerosol depressed surface temperature by 0.5-1.1 K globally. By contrast, one recent study (Feng and Zou, 2019) argued that aerosols contributed $+ 0.005 \pm 0.237$ K on global surface temperature change after 2000. Therefore, the aerosol effect on climate warming is still under discussion. Model simulations (Ji et al., 2015) have

shown that carbonaceous aerosols have positive radiative forcing effects on climate warming over the TP, leading to a 0.1–0.5 °C warming in the monsoon season, while some studies have demonstrated that anthropogenic aerosols (AAs) have a cooling effect on local climate warming (Smith et al., 2016;Sundström et al., 2015). Gao et al. (2015) contended that aerosols cool the

surface 0.8 – 2.8 K in North China Plain, and Li et al. (2015) calculated the aerosol impact on climate warming in different

seasons over arid-semiarid and humid-semiarid areas, and across China, and showed that China undergoes a cooling rate of

−0.86 to −0.76 °C per century due to increased aerosols. However, former conclusions were mainly based on model simulations

(Liao et al., 2015) and have not yet been combined with observations. Therefore, there is no consistent result of quantifying

the impact of anthropogenic aerosols on climate warming especially at the TP and the observations from multiple data sources

are urgently needed in the quantification.

In this study, we aim to analyze the long-term spatiotemporal variation of surface radiation over the TP by generating long-

term surface radiation datasets from satellite products and model simulations. Solar dimming is to be attributed by analyzing

multiple data sources. The depressing effect of aerosols on climate warming needs to be quantified in the end. Calibrated by

the Clouds and the Earth's Radiant Energy System Energy Balanced and Filled (CERES EBAF) Edition 4.0 surface downward

radiation products (Kato et al., 2018), long-term (from 1850–2015) surface DSR and DLR datasets over the TP were developed

by merging 18 Coupled Model Intercomparison Project Phase 5 (CMIP5) models (Taylor et al., 2012). Site validation and

comparison were processed to the calibrated data, the CMIP5 model outputs, and other long-term radiation products. The

spatiotemporal variations in the generated surface radiation datasets and four long-term air temperature datasets were initially

analyzed, and the TP solar dimming was attributed by using multiple types of satellite and reanalysis data, which were

confirmed by climate model simulations. We characterized the seasonal difference of the TP solar dimming and further

quantified the depressing effect on local climate warming since 1850 using two methods driven by satellite observations and

model simulations.

## 2 Data and Methodology

### 2.1 Data

#### 2.1.1 Coupled Model Intercomparison Project Phase 5 (CMIP5) data

The CMIP5 (Taylor et al., 2012) datasets with at least one dimension of spatial resolutions less than 2° were chosen in the

paper, and 18 monthly modeled datasets (summarized in Table 1) from the Historical experiment were included, which cover

1850 to 2005; the following years (2006–2015) of records are from the Representative Concentration Pathway (RCP) 8.5

experiment. We used the first ensemble (r1i1p1) only of each experiment to reduce the calibration complexity of surface

downward radiation. Aerosol optical depth (AOD), precipitation, and wind speed from the Historical (1850 - 2005) and RCP8.5

(2006 - 2015) experiments of the models were used to analyze differences in dimming magnitudes at seasonal scales (Figure

7). Surface temperature, wind speed, and relative humidity were employed for calculating the aerosol depressing effect due to

the long-term coverage. The corresponding HistoricalMisc experiment (i.e., an experiment that combined different specific

forcings) data were also utilized in the attribution analysis, including downward shortwave radiation driven by AA and noAA

(all forcings except AA). NoAA derived near-surface air temperature from multiple model ensembles is used in the depressing

effect analysis. The PiControl experiment provided the natural internal variation utilized in the optimal fingerprinting method (Methodology 2.2.2). The attribution and depressing effect calculation included all the available variable ensembles of each model, including wind speed, precipitation, surface temperature, air temperature, and relative humidity. All datasets were resampled into 1 Lat/Lon degree using a bilinear interpolation method. CMIP5 datasets are available from the Intergovernmental Panel on Climate Change (IPCC) data distribution center at the website (PCMDI, 2013).

### 2.1.2 Remote sensing and assimilation products

### 2.1.2.1 Clouds and the Earth's Radiant Energy System Energy Balanced and Filled (CERES EBAF) radiation products

The CERES EBAF-surface Edition 4.0 monthly downward shortwave and longwave radiation products (Kato et al., 2018) were employed as a benchmark for calibrating simulated CMIP5 surface radiation by a non-negative least square (NNLS) regression approach (Bro and De Jong, 1997) (Methodology 2.2.1). Comparing with former solar radiation products, CERES EBAF has been comprehensively assessed and considered as the most advanced surface radiation satellite product. It is usually used as a reference for model and reanalysis validation (Zhang et al., 2015;Zhang et al., 2016). It can capture the temporal variation of surface radiation by comparing it with surface measurements (Supplementary Fig. S1). Besides, former studies have already used the CERES EBAF DSR product for applications and analysis (Feng and Wang, 2018;Ma et al., 2015). This new version contains surface fluxes consistent with the top-of-atmosphere (TOA) fluxes provided from the CERES Energy Balanced and Filled Top of Atmosphere (EBAF-TOA) data product. They also used improved cloud properties that are corrected by cloud profiling radar, and consistent input sources are employed, such as temperature, humidity, and aerosol data, in order to solve the spurious anomaly problem (Jia et al., 2018;Jia et al., 2016). All the advantages help to quantify the absolute magnitude and temporal trend of surface DSR (Feng and Wang, 2019). The CERES EBAF-TOA Edition 4.0 monthly TOA albedo product (Loeb et al., 2018) was used for computing the temporal variation in the TOA albedo over the Tibetan Plateau (TP) in the dimming attribution analysis. The CERES EBAF TOA solar radiation and surface albedo products were used as a monthly climatology. TOA solar radiation was combined with the calibrated surface DSR data to compute the atmospheric shortwave transmissivity in the aerosol depressing effect quantification. Meta information on the included products is summarized in Table 2. All products were pre-processed into 1 degree at a monthly scale for further analysis to unify the spatial and temporal resolutions.

### 2.1.2.2 Aerosol, cloud, and dust products

Multiple aerosol, cloud, and dust products were employed for the attribution of solar dimming. The averages of the Moderate Resolution Imaging Spectroradiometer (MODIS) MOD/MYD08 Collection 6.1 aerosol optical depth (AOD) 550 products that combine the Dark Target and Deep Blue algorithms were used for detecting aerosol variations over the TP. The MOD/MYD08 C6.1 cloud fraction was also included in the attribution analysis. In MODIS C6.1, the brightness temperature biases and trending were substantially reduced compared to C6, which affected the cloud retrieval and also caused large uncertainty with respect to the AOD over elevated areas (Sogacheva et al., 2018). Additionally, the MODIS AOD coverage increased in C6.1.

The Sea-Viewing Wide Field-of-View Sensor (SeaWiFS) AOD550 product (Sayer et al., 2012) utilized 12 candidate aerosol models for generating the aerosol lookup tables (Hsu et al., 2012), and AODs at different wavelengths have been retrieved over land with the use of the Deep Blue algorithm (Pozzer et al., 2015). The SeaWiFS AOD550 product was used for calculating the aerosol temporal variation over the TP.

The aerosol index data were inversed from Total Ozone Mapping Spectrometer (TOMS)-Nimbus 7, TOMS-Probe, and Ozone Monitoring Instrument (OMI) at different time periods (Ahmad et al., 2006). We employed TOMS N7 records from 1980 to 1992, TOMS Probe from 1997 to 2004, and OMI from 2005 to 2015. By using the 340- and 380-nm wavelength channels (which have negligible dependence on ozone absorption), the aerosol index was defined based on backscattered radiance measured by TOMS and OMI and the radiance calculated from a radiative transfer model for a pure Rayleigh condition (Hsu

et al., 1999). This approach measures the relative amount of aerosols and has a comparable relationship with AOD (McPeters et al., 1998).

Particulate matter (PM) 2.5, characterizing very small particles that have a diameter of $< 2.5$ µm and are produced by human activities, is a common index of air pollution (Wang et al., 2015). We employed the PM2.5 satellite products to link the aerosol loading with air pollution. The PM2.5 satellite product is calculated from MODIS, Multi-angle Imaging SpectroRadiometer

(MISR), and SeaWiFS AOD products based on a relationship generated from a chemical transport model (Van Donkelaar et al., 2016), and its uncertainty is determined by ground measurements.

MERRA2 aerosol products are the new generation reanalysis data that have assimilated MODIS and MISR land aerosol products since 2000 (Randles et al., 2017). Dust column mass density from MERRA2 was included for comparison of the temporal variation with PM2.5 data to determine whether the AOD increase was due to air pollution.

International Satellite Cloud Climatology Project (ISCCP) and Pathfinder Atmospheres–Extended (PATMOS-X) provide long-term cloud fraction products, however, the trend is spurious (Evan et al., 2007). This is because of the satellite zenith angle, equatorial crossing time, and because the sensor calibration lacks long-term stability. Corrected cloud fraction datasets (Norris and Evan, 2015), which were used for solar dimming attribution in this paper, employed an empirical method for removing artifacts from the ISCCP and PATMOS-X, and the corrected cloud products have been used for providing evidence

for climate change in satellite cloud records in other studies (Norris et al., 2016).

The European Centre for Medium-Range Weather Forecasts reanalysis 5 (ERA5) (Hersbach and Dee, 2016), as the newest reanalysis dataset, is also employed for attributing the solar dimming. ERA5 is the fifth generation of ECMWF atmospheric reanalyses and follows the widely used ERA-Interim. By comparing with ERA-Interim, it has higher spatial and temporal resolutions and finer atmospheric levels. In addition, it includes various newly reprocessed datasets and recent instruments that could not be ingested in ERA-Interim. ERA5 provides atmospheric profiles with high accuracy by assimilating conventional

observations (e.g., balloon samples, buoy measurements) and satellite retrievals. Therefore, total column water vapor and cloud fraction from ERA5 are used in the study.

Diagnosing Earth's Energy Pathways in the Climate system (DEEP-C) TOA albedo was calculated by the DEEP-C monthly TOA absorbed solar radiation (ASR) (Allan et al., 2014) and monthly TOA incoming solar radiation climatology from the CERES EBAF-TOA Ed4.0 product. The DEEP-C TOA albedo was used for detecting the radiative influence of increasing aerosols in the dimming attribution. Central to the DEEP-C TOA radiation reconstruction are monthly observations of the TOA radiation from the CERES scanning instruments after 2000 and Earth Radiation Budget Satellite (ERBS) wide field of view (WFOV) nonscanning instrument from 1985 to 1999. A strategy was required to homogenize the satellite datasets (Allan et al., 2014), and the ERA-Interim has provided atmospheric information since 1979, also using a subset of nine climate models to represent direct and indirect aerosol radiative forcings.

**2.1.2.3 Surface albedo products**

According to He et al. (2014), the fine-resolution (0.05°) climatological surface albedo products retrieved from satellite observations agree well with each other for all the land cover types in middle to low latitudes. Therefore, we selected four commonly used satellite surface albedo products for calculating the surface albedo climatology over the TP, including the CERES EBAF, the Global LAnd Surface Satellite (GLASS), the Clouds, Albedo, and Radiation-Surface Albedo (CLARA-SAL), and the GlobAlbedo from 2001 to 2011 (covered by the four products). First, we generated the monthly climatological albedo of each satellite product, and we computed all standard deviations of any possible three product climatology combinations at each pixel. Then for each pixel, we chose the product combination that has the lowest standard deviation and calculated the mean value to represent the ground truth climatology.

The GLASS albedo product from MODIS observations is based on two direct albedo estimation algorithms (He et al., 2014); one designed for surface reflectance and one for TOA reflectance (Qu et al., 2014). The statistics-based temporal filtering fusion algorithm is used to integrate these two albedo products (Liu et al., 2013a). The GLASS albedo product has been assessed by ground measurements and the MODIS albedo product and has a comparable accuracy (Liu et al., 2013b). The CLARA-SAL product is inversed from advanced very high resolution radiometer (AVHRR) observations (Riihelä et al., 2013). Atmospheric correction was done by assuming AOD and ozone is constant. Sensor calibration and orbital drift have been dealt with and the uncertainty of monthly albedo estimation is about 11%. The GlobAlbedo product uses an optimal estimation approach European satellites, including Advanced Along-Track Scanning Radiometer (AATSR), SPOT4-VEGETATION, SPOT5-VEGETATION2, and Medium-Resolution Imaging Spectrometer (MERIS) (Lewis et al., 2013). MODIS surface anisotropy information was used for gap-filling. More detailed algorithm introductions and comparison can be found in He et al. (2014).

**2.1.2.4 Advanced Spaceborne Thermal Emission and Reflection Radiometer Global Monthly Emissivity Dataset (ASTER_GED)**

The ASTER surface emissivity data were also used for calculating the aerosol depressing effect. The ASTER_GED data products are generated using the ASTER Temperature Emissivity Separation algorithm (Gillespie et al., 1998) atmospheric

correction method. This algorithm uses MODIS Atmospheric Profiles product MOD07 and the MODTRAN 5.2 radiative transfer model, snow cover data from the standard monthly MODIS/Terra snow cover monthly global 0.05° product MOD10CM, and vegetation information from the MODIS monthly gridded NDVI product MOD13C2 (NASA JPL. ASTER Global Emissivity Dataset, 2016). Surface broadband emissivity is calculated according to Cheng et al. (2014).

### 2.1.3 Ground measurements

Utilized for surface radiation validation, ground radiation measurement sites over the TP are mainly from two ground networks (Global Energy and Water Exchanges [GEWEX] Asian Monsoon Experiment [GAME] (Yasunari, 1994) and Coordinated Energy and Water Cycle Observation Project (CEOP) Asia-Australia Monsoon Project [CAMP] (Leese, 2001)) that cover 1995–2005. The GAME was proposed as an international project under the Global Energy and Water Exchanges (GEWEX) program to understand the processes associated with the energy and hydrological cycle of the Asian monsoon system, and its

variability. The data are available at GAME-ANN (2005). The CAMP, which followed the GAME, focused on water and energy fluxes and reservoirs over specific land areas and monsoonal circulations. These data are available at CAMPTibet (2006). We ignored the spatial representative difference between site observations and downward radiation datasets in line with former studies (Wang and Dickinson, 2013;Zhang et al., 2015).

We also collected ground observations from 5 China Meteorological Administration (CMA) sites over the TP to detect long-

term temporal variation of surface DSR from 1958 to 1980. The observations after 1980s are not included due to the data discontinuity issue (Zhang et al., 2015). Even the site amount is not large enough to represent the whole TP, the sampled surface measurements can reflect the ground truth and prove the dimming over the TP (Supplementary Fig. S2). 22 sites that observed downward radiation in the TP were included in this study, and their distribution is shown in Figure 1. The Tibetan Plateau region is defined as the Chinese Qinghai-Tibet Plateau in this paper, covering most of the Tibet Autonomous Region

and Qinghai in western China (Wang et al., 2016).

### 2.1.4 Surface air temperature datasets

Four long-term surface air temperature datasets were used for characterizing the temporal variation over the TP and corresponding aerosol depressing effect; the Berkley Earth Surface Temperature land surface air temperature dataset (BEST-LAND) (Rohde et al., 2013), Climate Research Unit Temperature Data Set version 4 (CRU-TEM4v) (Jones et al., 2012),

National Aeronautics and Space Administration Goddard Institute for Space Studies (NASA-GISS) (Hansen et al., 2010), and National Oceanic and Atmospheric Administration National Center for Environmental Information (NOAA-NCEI) (Smith et al., 2008). These data were interpolated and homogenized from ground observation networks. Rao et al. (2018) provided data assessment of the four air temperature datasets, and a brief summary of the datasets is provided in Supplementary Table S1.

**2.2 Methodology**

**2.2.1 Calibration method**

Long-term series surface downward radiation data are urgently needed for characterizing the spatial-temporal variation of surface radiation budget over the TP, and they are also essential for the solar dimming attribution and calculating the aerosol radiative forcing and depressing effect on the increasing air temperature. To generate a long-term series surface downward radiation data record with high accuracy, a non-negative least squares (NNLS) regression (Bro and De Jong, 1997) approach

was employed for merging multiple CMIP5 model records. The non-negative multiple linear models are utilized because the non-negativity constraint of NNLS only applies a non-subtractive combination of all components (Eq. [1]).

$$y = \sum a_i x_i , (a_i \geq 0), \tag{1}$$

where $y$ is the calibrated radiation data, $a_i$ is the coefficient of each CMIP5 model simulation $x_i$.

The calibration was done pixel by pixel to avoid the influence of geolocation and elevation. Given that downward radiative

fluxes are always positive, a key assumption here is that the CERES radiative flux can be expressed as a non-negative linear combination of each model output at each grid. One CMIP5 model may hardly present the variations in the actual radiative fluxes but should not have any negative contributions. To avoid the influence from the seasonal cycle on the expression of the inter-annual variation of the radiative flux, the fusion models were generated monthly. The CERES satellite products aid NNLS to provide the best-weighted combination for each CMIP5 model, producing improved validation results compared to those

produced by only using the mean of all model outputs. Mean Bias Error (MBE), Root-Mean-Square-Error (RMSE), and $R^2$ were utilized for quantifying site validation and comparing with the mean CMIP5 data and multiple satellite and reanalysis product fused radiation data from Shi and Liang (2013). The temporal variation and comparison among products are calculated by using latitude-weighted average over the TP. A detailed description of assessment methods is introduced in Jia et al. (2018).

**2.2.2 Attribution analysis**

Optimal fingerprinting is a common method in the attribution analysis of model data. It is based on a linear relationship, and, if the scaling factor is > 0 at a certain significance level, the variable has a positive contribution toward the responding variable. The optimal fingerprinting method has been widely applied for climate change detection and attribution (Sun et al., 2014;Zhou et al., 2018). It is assumed that the response variable has a linear relationship with different driving variables (Eq. [2]):

$$y = \sum \beta_i x_i + \varepsilon, \tag{2}$$

where $\varepsilon$ is the modeled natural internal variation obtained from the CMIP5 piControl Experiment, and $\beta_i$ is the scaling factor of each driving variable. $x_i$ are the DSR simulation results from averages of aerosol-driven experiment ensembles and non-aerosol-driven experiment ensembles in this study. If the scaling factor is > 0 at a certain significance level (in this study, 5% – 95% uncertainties are estimated based on Monte Carlo simulations), the variable has a positive contribution toward the

responding variable. In CMIP5 HistoricalMisc experiment, anthropogenic forcings are focused, thus in this study only experiment with anthropogenic aerosols (AA) and without AA (noAA) were employed for the attribution. Impact simulations of cloud/water vapor are not covered in the experiment and we assumed that noAA experiment can represent the model simulation about the cloud/water vapor impacts on surface downward shortwave radiation.

### 2.2.3 Depressing effect

For quantitating the depressing effect of aerosols on surface warming over the TP, the shortwave and longwave radiative effects must be separated. To calculate the relationship between the surface air temperature increase and surface radiation components, it is necessary to decompose the energy sources into separate mechanisms. The land surface energy balance is given by Eq. (3):

$$S_n + L_n = \lambda E + H + G, \tag{3}$$

where $S_n$ is the net shortwave radiation, $L_n$ is the net longwave radiation, $\lambda$ is the latent heat of vaporization, $E$ is evapotranspiration, $H$ is the sensible heat flux, and $G$ is the ground heat flux that was neglected for the long time period. They can be decomposed in Eq. (4) as:

$$S\tau(1-\alpha) + \varepsilon_s\sigma(\varepsilon_a T_a^4 - T_s^4) = \lambda E + \rho C_d (T_s - T_a)/r_a, \tag{4}$$

where $S$ is the TOA solar radiation, $\tau$ is the atmospheric shortwave transmissivity (ratio between the calibrated surface DSR and S), $\alpha$ is the surface albedo, $\varepsilon_s$ is the surface broadband emissivity, $T_a$ is the air temperature, $T_s$ is the surface skin temperature, and $r_a$ is the aerodynamic resistance at 2 m height. $\sigma$ is equal to $5.67 \times 10^{-8}\,\mathrm{W\,m^{-2}\,K^{-4}}$, $\rho$ is 1.21 kg m$^{-3}$, and $C_d$ is 1013 J kg$^{-1}$ K$^{-1}$. $\varepsilon_a$ is the air emissivity parameterized based on Carmona et al. (2014) that has the highest accuracy by comparing with other parameterization methods (Guo et al., 2019). Considering that the $\Delta T_a$ is mainly dominated by the change in $T_s$ interacting with $T_a$ through radiative and thermal processes and the change in atmospheric circulation ($\Delta T_a^{cir}$, for example, advection of cold and warm air masses), a first-order approximation of the direct near-surface temperature response to each component (Zeng et al., 2017) is derived from Eq. (5) and Eq. (6):

$$\Delta T_a = 1/f\, (S(1-\alpha)\Delta\tau - S\tau\Delta\alpha - \lambda E + \varepsilon_s\sigma T_a^4\Delta\varepsilon_a$$
$$+ \rho C_d((T_s - T_a)/r_a^2)\Delta r_a) + \Delta T_a^{cir}, \tag{5}$$

where the $f$ is:

$$f = \rho C_d/r_a + 4\varepsilon_s\sigma\varepsilon_a T_a^3, \tag{6}$$

and $f^{-1}$ represents the land surface air temperature sensitivity to 1 W m$^{-2}$ radiative forcing at the land surface. We assumed that $S$, $\lambda$, $\rho$, $C_d$, $\sigma$, and $\varepsilon_s$ are independent of $T_s$. We employed the $\alpha$, $\varepsilon_s$, and $S$ climatologies and the mean values of the satellite products for several years (details in Table 2). Therefore, the $T_a$ response to the surface DSR change is calculated by the first

term (related to $\Delta\tau$) in the formula. $\Delta T_a^{cir}$ are not computed in the analysis because they have limited impact on aerosol radiative effect. Based on Eq. (7):

$$\text{DLR} = \sigma\varepsilon_a T_a^4, \tag{7}$$

from Eq. (8) we can also obtain the relationship between $\Delta T_a$ and $\Delta\text{DLR}$:

$$\Delta T_a^{DLR} = \Delta\text{DLR}/(4\sigma\varepsilon_a T_a^3). \tag{8}$$

We then added the depressing effect from $\Delta\text{DSR}$ and $\Delta\text{DLR}$ to get the aerosol depressing effect on the TP climate warming. The first-order approximation method included many reliable remote sensing products and it's reasonable to use climatology of the TOA and surface variables in the equation (Supplementary Fig. S8). We also calculated the depressing effect of AAs by employing the CMIP5 air temperature data from multiple noAA simulations, which used physical parametrization methods. The two methods can validate each other. We then compared the two results (Supplementary Fig. S7) and calculated the mean value as the final depressing effect result.

## 3 Results and discussion

### 3.1 Validation and comparison of the calibrated downward radiation data

From January 1995 to December 2005, in situ observations were collected at 17 sites for monthly validation of DSR and DLR. The scatter diagrams and validation results of the CERES calibrated data, mean CMIP5, and reanalysis and satellite fused product from Shi and Liang (2013) at two networks are shown in Figure 2.

Figure 2 indicates that the CERES calibrated datasets have the lowest bias and RMSE for DSR and DLR validation at GAME and CAMP network, and the $R^2$ at CAMP network has the highest. The bias of the calibrated DSR at CAMP (GAME) is -0.27 (-3.68) Wm$^{-2}$ and the RMSE is 20.59 (25.27) Wm$^{-2}$, whereas the bias of calibrated DLR at CAMP (GAME) is 0.63 (-4.31) Wm$^{-2}$ and the RMSE is 11.90 (21.08) Wm$^{-2}$. We can conclude that by using the NNLS method, the CERES calibration decreased the data bias and RMSE and improved the $R^2$, providing the best-weighted combination for each CMIP5 model and producing better validation results than those produced by only using the mean of all model outputs and data from former studies. The minor validation RMSE difference (4.68 Wm$^{-2}$ in DSR and 9.18 Wm$^{-2}$ in DLR) between the two networks is mainly caused by disparate instruments and different site numbers. We ignored the spatial mismatch between site observations and downward radiation datasets in line with former studies (Wang and Dickinson, 2013;Zhang et al., 2015). Moreover, the annual anomaly temporal variation was compared with the data from Shi and Liang (2013) and is shown in Figure 3.

The annual temporal variation illustrates that two products have a similar temporal trend of DSR and DLR at a decadal scale over the TP, and that the CERES calibrated data have longer time spans. The DSR output shown by Shi and Liang (2013) has a slightly larger decreasing trend, which may be because GEWEX and ISCCP surface radiation products have high weights in

the model, and the related spurious cloud trend causes an overestimated dimming trend (Evan et al., 2007). Feng and Wang (2018) utilized a cumulative probability density function-based method to fuse CERES and longer time reanalysis surface DSR data, however in this study they only calibrated individual reanalysis whereas in the present study we merged multiple CMIP5 data that cover longer time span and include more climate general model simulations. Besides, most reanalysis did not include aerosol variation information, which is important to determine the DSR decadal variation and characterize the aerosol radiative forcing.

## 3.2 Characterizing long-term variations in air temperature and surface downward radiation over the Tibetan Plateau

Air temperatures slowly increased from 1850 to 2015. The DLR increased gradually prior to 1970 but rapidly during the following period. The DSR remained stable and decreased rapidly afterward, revealing an opposite trend to the DLR and air temperature (Figure 4a). In total, DSR decreased by 4.1 W m$^{-2}$ from 1850 to 2015 with a gradient of -0.53 W m$^{-2}$ per decade after 1950, and DLR increased from 0.21 W m$^{-2}$ per decade to 1.52 W m$^{-2}$ per decade after 1970. Air temperature has increased by 1.39 K since 1850. Prior to 1950, increased air temperature was mainly triggered by increased DLR. Air temperature slightly decreased from 1950 to 1970, because both DSR and DLR decreased during that period. Although DLR increased rapidly after 1970, the air temperature gradient has not considerably changed, mainly due to solar dimming diminishing the greenhouse effect. The solar dimming over the TP is also detected by long-term ground DSR observations from 5 CMA sites since 1958 (Supplementary Fig. S2) when the measurements were set up. Because heat exchange with other regions in the summer and winter are mostly counteracted, we ignored the interaction at decadal scales and suggested that air temperature is mainly driven by local radiation components. All four surface air temperature datasets show similar temporal trends, especially on the decadal scale.

The long-term spatiotemporal variations in downward surface radiation over the TP and surrounding regions are illustrated in Figure 4b and 4c. Figure 4b demonstrates that the DSR decrease rate in the central region is about -0.08 W m$^{-2}$ per decade, much lower than in surrounding areas. The fastest decrease in DSR appears in the southeastern TP at a gradient of about -0.37 W m$^{-2}$ per decade since 1850. Northern India, South Asia, and southern China show a substantially dimming trend with a gradient of about -0.65 W m$^{-2}$ per decade. The DLR has increased, especially in the central and northern TP (Figure 4c). However, the rate of increase is much slower in the southern and southeastern TP, with a gradient of approximately 0.21 W m$^{-2}$ per decade.

### 3.3 Attribution to the Tibetan Plateau solar dimming

### 3.3.1 Analysis of satellite products and reanalysis

Both observed satellite products and reanalysis datasets provide evidence that AAs are the major driver of the significant decrease in the DSR over the TP. The aerosol optical depth (AOD) products from the SeaWiFS (Sayer et al., 2012) and MODIS 08 C6.1 (Levy et al., 2007) satellite products demonstrated similar annual anomalies and slightly increasing trends after 1998

(Figure 5a). The aerosol index, which measures the relative amount of aerosols and has a comparable relationship with AOD (McPeters et al., 1998), shows that the number of aerosols has been escalating over the past 30 years (Figure 5a). The AOD has increased 0.0098 over the TP based on the trend, causing approximately 1.97 Wm$^{-2}$ of dimming since 1998, according to multiple CMIP5 AA simulations over the TP and near linear relationship between AOD and radiative forcing at this magnitude level (Yang et al., 2012). This is larger than the calibrated DSR dimming result of 1.10 Wm$^{-2}$, and we inferred that some of the dimming caused by the aerosol increase is offset by decreased cloud cover (Figure 5c).

The PM2.5 satellite product (Van Donkelaar et al., 2015) also showed an increasing trend after 2000 (Figure 5b), while MERRA2 dust loading (Randles et al., 2017), which has been assimilated from MODIS and MISR land aerosol products since 2000, has been decreasing. Particulate matter (PM) 2.5, characterizing very small particles that have a diameter of less than 2.5 micrometers and are produced by human activities, is a common index for measuring air pollution (Wang et al., 2015). The variation in PM2.5 and dust indicates that increased aerosols are mainly from air pollution rather than from natural causes.

Based on the corrected cloud fraction datasets (Norris and Evan, 2015) from ERA5, the Pathfinder Atmospheres–Extended (PATMOS-X) and the average cloud fraction from MOD/MYD08 C6.1, the results demonstrate that the cloud fraction over the TP has been decreasing since 1980 (Figure 5c), indicating a trend opposite to the TP solar dimming. The temporal variation of ISCCP and ERA5 is stable with a larger $p$ value than 0.05. Even different long-term products have uncertainties especially before 1990, the overall variation is decreasing after 1990. Therefore, we inferred that the overall trend demonstrated that cloud coverage is not a dimming driver. Moreover, former studies also found the decreasing cloud coverage trend based on site observations since 1960s (Kuang and Jiao, 2016;Yang et al., 2012). The overall temporal variation of TOA albedo from DEEP-C (Allan et al., 2014) and CERES presents an increasing trend with a magnitude of ~0.01 over the TP from 1985 to 2015 (Figure 5d). The TOA albedo is an important component of Earth's energy budget and is mainly influenced by clouds and aerosols. It can be inferred that aerosols over the TP were recently increasing, reflecting more solar radiation into space and causing the TOA albedo increase and solar dimming at the surface.

Yang et al. (2012) argued that aerosols had limited radiative forcing compared with the DSR decrease over the TP, however, ground observations used in these studies were only from one AErosol RObotic NETwork (AERONET) site at the center of the TP (30.773 °N, 90.962 °E). Furthermore, the site was less impacted by surrounding regions and so cannot represent the entire TP, especially the edge regions that are more easily contaminated by air pollution from surrounding areas (Cao et al., 2010).

Yang et al. (2012) and (2014) also suggested that the increasing trend in the deep convective clouds were attributed to the solar dimming, however, our analysis has a different conclusion. We used the cloud top pressure and cloud optical depth released by MODIS at different atmospheric levels to detect which pixels are deep convective clouds. The threshold is based on the definition of ISCCP (Rossow and Schiffer, 1999). Then, we calculated the ratio of deep convective clouds in all pixels over the TP and obtained the temporal variation and spatial distribution. Moreover, regardless of cloud type, the cloud mainly affects

the DSR by cloud optical depth, so we reviewed the temporal variation of the total cloud optical depth. The analysis illustrates that even though the ratio of deep convective clouds is increasing (Supplementary Fig. S3a), deep convective clouds only appeared in the south and west of the TP (Supplementary Fig. S3b). Additionally, the overall cloud optical depth has been slightly decreasing over the past 15 years (Supplementary Fig. S3a). Therefore, deep convective clouds have little influence over the entire TP.

By analyzing MODIS satellite products and ERA5 reanalysis, we also assessed the temporal variation of the atmospheric water vapor and total column water vapor (Supplementary Fig. S4), suggested as an important driving factor of the solar dimming by Yang et al. (2012), and the temporal variation illustrated that water vapor has been considerably decreasing since 1998 over the TP. However, solar dimming did not show a similar turning point around 1998 in their site observations and our results, and rather the overall increasing trend of water vapor since 1980 was limited. Therefore, the influence of water vapor variations

can be ignored. The Yang et al. (2012) study also identified this phenomenon by using ECMWF Re-Analysis (ERA)-40, however, the analysis result ceased in 2005 and did not show an overall turning trend.

    Although satellite products analysis only began in 1980, observational records have existed for more than 30 years, and the spatial extent of the satellite data over the entire TP supports our conclusions. Figure 4b shows that the regions of China, India, and South Asia surrounding the TP have large populations and serious air pollution. Balloon-borne observations (Tobo et al.,

2007) and remote sensing products (Vernier et al., 2015) have shown that fine aerosols can be transported over the TP region and enter the Asian tropopause aerosol layer by deep convection via two key pathways over heavily polluted regions (Lau et al., 2018). Moreover, other studies also reported that black carbon deposition altered surface snow albedo and accelerated melting in the TP mountain ranges (Qian et al., 2015). Because of the decreasing dust amount trend the TP, we infer that the increased aerosols are mainly due to air pollution around the TP. Although the TP is still one of the cleanest areas in the world

and the aerosol climatology is low, the dimming can be demonstrated by the variation of DSR decadal anomalies. It is necessary to point out that direct radiative effects (scattering and absorption effect) play a dominant role in the interactions between aerosols and the atmosphere (Li et al., 2017) when the aerosol loading is low. Thus the TP is easily affected by the aerosols increase under a clean atmosphere condition.

### 3.3.2 Analysis of model simulations

The long-term model simulation results also show that AAs are the main driving factor of solar dimming. Based on the multiple CMIP5 HistoricalMisc (an experiment combining different specific forcings) model ensembles, we found that the calibrated DSR and the DSR driven by AAs and noAA had stable variations before 1950. The calibrated DSR obviously decreased after 1950 (Figure 6a). Only the AA-driven DSR can capture the dimming trend since 1950, therefore we can conclude that AAs are the main signal at the decadal scale, while the factors in the noAA-driven model process (such as cloud cover and water

vapor) can be ignored. We also detected the AA and noAA driving factors using the optimal fingerprint method, and observed that AAs had a positive contribution, especially after 1970 (Figure 6b) when AA and historical data showed considerable

decrease while noAA kept stable. The impact factor of noAA is negative, and the satellite cloud products reveal the same conclusion after 1980 (Figure 5c), i.e., that the cloud fraction has been decreasing and has had a negative contribution to the TP solar dimming. DSR was driven by more forcings in noAA than AA experiments, introducing more uncertainties among models simulations after 1970. Therefore, it is possible that noAA impact factor shows a larger uncertainty bar. We inferred that cloud coverage dominated the negative natural impact because water vapor is quite stable since 1980s (Supplementary Fig. 4) that had little impact on the dimming.

General climate models have coarser spatial resolution, causing a lower elevation in the model than in reality and this may cause higher AOD estimation over highland area. However, when we compared the multiple CMIP5 AOD with site measurements, it demonstrates that the overall magnitude and monthly variation of CMIP5 AOD match the AERONET observations (Supplementary Fig. S5), even though it is slightly higher than the AOD in the non-monsoon season. Therefore, it is reasonable to include the CMIP5 AA and noAA simulations in the attribution work. Results of multiple models have uncertainties that are illustrated as colored shadow (standard deviation at each year) and the impact factor of analysis in 1970 – 2005 passed the significance test.

### 3.4 Depressing effects of aerosols on climate warming in summer

By comparing the first 30 years of climatology (1850–1880) and the last 30 years of climatology (1985–2015), we found that the TP solar dimming is stronger in summer (from June to August), at the same time that the increasing magnitude of the surface air temperature is the smallest (Figure 7a). Multiple CMIP5 model ensembles show that changes in precipitation and wind speed over the TP during different seasons were related to the AOD increase. Precipitation in summer had the greatest decrease relative to other seasons, while AOD increased more in the summer (Figure 7b). Furthermore, the wind speed clearly decreased in summer compared to other seasons (Figure 7c). The TP is a strong heat source in summer, forming a sensible heat–derived air-pump that dominates the atmospheric circulation (Wu et al., 2015) and conveys aerosols into the lower stratosphere (Lau et al., 2018). However, increased precipitation will reduce the aerosol duration lifetime considerably (Liao et al., 2015), and wind speed also controls aerosol diffusion. Hence it is evident that less precipitation and lower wind speeds in summer resulted in greater aerosol stability.

Although aerosols have considerably influenced summer downward radiation over the TP, their radiative effect on climate warming has not been quantitatively calculated. By employing the multiple noAA model simulations, the temporal variation of aerosol radiative forcing over the TP are illustrated in the Supplementary Fig. S6. It demonstrates that the aerosol radiative forcing has been increasing about 8.08 Wm$^{-2}$ by calculating the difference between the first 30 years of climatology (1850–1880) and the last 30 years of climatology (1985–2015).

Quantification of the depressing effect from AAs is essential for evaluating the impact of air pollution on local and continental climate warming and is also vital for improving our understanding of the role of human activities in remote areas. The depressing effects of aerosols on air temperature in summer (Figure 8) were calculated using two methods: one is using first-

order approximations of the direct near-surface air temperature response to each radiative and thermodynamic component and is based on remote sensing and modeling data; the other is calculated by using multiple noAA simulations. The two methods had similar depressing magnitudes (Supplementary Fig. S7), and the mean is shown in Figure 8. Surface air temperature increased almost 0.86 K (Figure 7a) over the TP in summer when comparing the first 30 years and the last 30 years, whereas the increasing magnitude of the surface air temperature that has no aerosol impact (the red line in Figure 8) is approximately $1.64 \pm 0.28$ K, which indicates that approximately $0.80 \pm 0.28$ K ($48.6 \pm 17.3\%$) of the local climate warming over the TP has been depressed by aerosols since 1850 in summer.

The first-order approximation method utilized many remote sensing products as climatology and forcing input, which are more reliable than the model simulation input (Supplementary Fig. 8), while we also calculated the depressing effect of AAs by employing the CMIP5 air temperature data from multiple noAA simulations that used physical parameterization methods. Two methods can validate with each other. Then we calculated the mean value as the final depressing effect result for including the advantages of these two methods.

The attribution of solar dimming over the TP and corresponding aerosol effect quantification revealed that anthropogenic aerosols dominate the solar radiation decrease and depress the climate warming in recent decades. Aerosols are cloud condensation nuclei (CCN) and more CCN may depress the cloud formation and precipitation. Moreover, the black carbon aerosol deposition may affect snow albedo feedback (Qian et al., 2015). Thus future studies need to analyze the indirect effect of aerosol loading (Qian et al., 2015) over there. However, it should be noticed that we don't conclude that the TP undergoes warming mitigation. In fact, the TP has a rapid warming rate than global warming (Yao et al., 2018) and other varying factors also affect the warming rate, in terms of the water vapor variation around 1998 (Supplementary Fig. S4). Water vapor is a weak DSR-absorbing factor but major greenhouse gas emitting downward longwave radiation, so its decrease might slow down the local warming rate. However, the impact of water vapor variation after 1998 is at an annual scale that cannot match the analysis in this study, thus follow-up researches may focus on it.

**4 Conclusions**

The TP plays a vital role in regional and global climate change due to its location and orography. Former studies have proven that this region undergoes significant climate change, however, the causes and impacts of solar dimming are still under debate. Calibrated by the CERES EBAF surface downward radiation products and using NNLS method, long-term (from 1850–2015) surface DSR and DLR datasets over the TP were developed by merging 18 CMIP5 models. Compared with the mean of multiple CMIP5 data and fusion data from former studies, the CERES calibrated data had the lowest bias and RMSE for DSR and DLR validation at GAME and CAMP network, and the highest $R^2$ at CAMP network. The calibrated DSR and DLR have similar temporal trends over the TP at a decadal scale compared to the fusion of multiple reanalysis and satellite products.

Based on calibrated surface downward radiation data and four sets of air temperature data, we characterized the spatiotemporal variation in surface radiation along with air temperature. The TP is currently experiencing substantial climate warming and solar dimming at the surface. In total, DSR decreased by 4.1 W m$^{-2}$ from 1850 to 2015 with a gradient of -0.53 W m$^{-2}$ per decade after 1950, and DLR increased from 0.21 W m$^{-2}$ per decade to 1.52 W m$^{-2}$ per decade after 1970. Air temperature has increased by 1.39 K since 1850. The dimming is also detected from long-term observing CMA sites. Spatial and temporal analyses illustrated that the DSR decrease rate in the central region was approximately -0.08 W m$^{-2}$ per decade, much lower than in surrounding areas. The fastest decrease in DSR appeared in the southeastern TP at a gradient of about -0.37 W m$^{-2}$ per decade since 1850, and DLR has increased, especially in the central and northern TP. However, the rate of increase is much slower in the southern and southeastern TP, with gradients of approximately 0.21 W m$^{-2}$ per decade.

By employing satellite and reanalysis products of aerosols, PM2.5, dust, cloud fractions, and TOA albedo, we determined that anthropogenic aerosols were the main cause of the solar dimming over the TP. The aerosol optical depth and the aerosol index has increased since the 1980s over the TP and increasing PM2.5 and decreasing dust linked the increasing aerosol to air pollution. We also proved from satellite products and reanalysis data that deep convective cloud and atmospheric water vapor are not the main drivers, due to limited distribution and magnitude since the 1980s. Furthermore, the overall cloud optical depth is decreasing. Additional evidence from multiple CMIP5 HistoricalMisc experiment ensembles also supports this conclusion that anthropogenic aerosols were the main cause of solar dimming over the TP.

Solar dimming over the TP is stronger in summer when the increasing magnitude of the surface air temperature is the smallest. Decreased precipitation and wind speeds triggered increased aerosol stability. Comparing the averages of the first 30 years (1850–1880) and last 30 years (1985–2015), the surface air temperature increased by approximately 0.86 K over the TP in the summer. The depressing effect of aerosol was calculated using two methods and both of which showed similar depressing magnitude. The increasing magnitude of the surface air temperature (with no aerosol impact) was approximately 1.58 K, which means approximately 0.80 ± 0.28 K (48.6 ± 17.3%) of the local climate warming over the TP has been depressed by aerosols from 1850 to 2015 in summer. The study reveals the impacts of human activities on regional warming, even in remote areas, and highlights the need for additional studies to be conducted to quantify the influence of air pollution on regional climate change over the TP. We will focus on the influences of air pollution on local precipitation over the TP and surrounding areas in the next work.

**Author contributions**

S. Liang conceived and scoped the research. A. Jia, B. Jiang, and X. Zhang downloaded and pre-processed the data. A. Jia and D. Wang performed data statistics and the interpretation of the results. A. Jia and S. Liang wrote the manuscript. All authors contributed to revising the article.

## Acknowledgments

This study was supported by NASA under grant 80NSSC18K0620 and the Chinese Grand Research Program on Climate Change and Response (projects 2016YFA0600101 and 2016YFA0600103). We gratefully acknowledge the Intergovernmental Panel on Climate Change (IPCC) data distribution center for providing the model simulation outputs. We also thank the CERES science team, MODIS science team, Goddard Earth Sciences Data and Information Services Center (GES DISC), Earth Observing System Data and Information System (EOSDIS), Atmospheric Composition Analysis Group at Dalhousie University, Research Data Archive at the National Center for Atmospheric Research (NCAR), GLASS science team, and Land Processes Distributed Active Archive Center (LP DAAC) for providing satellite and reanalysis products. We acknowledge the Berkeley Earth, Goddard Institute for Space Studies, National Centers for Environmental Information, and Climate Research Unit for providing near-surface air temperature datasets. We thank the GEWEX Asian Monsoon Experiment (GAME), CEOP Asia-Australia Monsoon Project, and Aerosol Robotic Network for providing ground measurements. We also thank the contributions of the referees and one reader during the open discussion.

## Data and code availability

The authors declare that the data we generated will be available after acceptance. All datasets supporting the findings of this study were identified by referring citations in the references section. Analysis scripts are available by request to S. Liang.

## Competing financial interests

The authors declare no competing financial interests.

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

**Table 1: Summary of the Coupled Model Intercomparison Project Phase 5 (CMIP5) surface downward radiation simulations used in this study.**

| Name | Spatial Resolution | | Reference |
|:---:|:---:|:---:|:---:|
| | Longitude | Latitude | |
| CMCC-CM | 0.75° | 0.75° | Scoccimarro et al. (2011) |
| CESM1-CAM5 | 1.25° | 0.94° | Meehl et al. (2013) |
| CESM1-BGC | 1.25° | 0.94° | Long et al. (2013) |
| CCSM4 | 1.25° | 0.94° | Gent et al. (2011) |
| MRI-CGCM3 | 1.13° | 1.13° | Yukimoto et al. (2012) |
| BCC-CSM1.1m | 1.13° | 1.13° | Wu et al. (2010) |
| MIROC5 | 1.41° | 1.41° | Mochizuki et al. (2012) |
| CNRM-CM5 | 1.41° | 1.41° | Voldoire et al. (2013) |
| ACCESS1.0 | 1.88° | 1.24° | Franklin et al. (2013) |
| ACCESS1.3 | 1.88° | 1.24° | Franklin et al. (2013) |
| IPSL-CM5A-MR | 2.50° | 1.26° | Dufresne et al. (2013) |
| INMCM4 | 2.00° | 1.50° | Volodin et al. (2010) |
| MPI-ESM-LR | 1.88° | 1.88° | Jungclaus et al. (2010) |
| MPI-ESM-MR | 1.88° | 1.88° | Jungclaus et al. (2010) |
| CSIRO-Mk3.6.0 | 1.88° | 1.88° | Gordon et al. (2010) |
| CMCC-CMS | 1.88° | 1.88° | Scoccimarro et al. (2011) |
| NorESM1-M | 2.50° | 1.88° | Tjiputra et al. (2013) |
| NorESM1-ME | 2.50° | 1.88° | Tjiputra et al. (2013) |

**Table 2: Meta information on the satellite and reanalysis products. All products were resampled into 1 Lat/Lon degree using bilinear interpolation or spatial averaging in the paper. All data were accessed on 15 December 2018.**

| Variable | Version | Time Span | Spatial Resolution | Data Availability | Usage |
|---|---|---|---|---|---|
| **DSR** | CERES EBAF-surface Ed4.0 | 2001.01–2015.12 | 1°×1° | https://ceres.larc.nasa.gov/order_data.php | calibration |
| **DLR** | CERES EBAF-surface Ed4.0 | 2001.01–2015.12 | 1°×1° | https://ceres.larc.nasa.gov/order_data.php | calibration |
| **TOA albedo** | CERES TOA-surface Ed4.0 | 2001.01–2015.12 | 1°×1° | https://ceres.larc.nasa.gov/order_data.php | attribution & depressing effect |
| **AOD** | MOD/MYD08 C6.1 | 2001.01–2015.12 | 1°×1° | https://earthengine.google.com/ | attribution |
| **cloud fraction** | MOD/MYD08 C6.1 | 2001.01–2015.12 | 1°×1° | https://earthengine.google.com/ | attribution |
| **atmospheric water vapor** | MOD/MYD08 C6.1 | 2001.01–2015.12 | 1°×1° | https://earthengine.google.com/ | attribution |
| **AOD** | SeaWIFS 1.0_L3M | 1998.01–2010.12 | 1°×1° | https://disc.gsfc.nasa.gov/datasets?page=1 | attribution |
| **aerosol index** | TOMS & OMI Aerosol Index L3 | 1980.01–1993.12, 1997.01–2015.12 | 1°×1.25° | https://disc.gsfc.nasa.gov/datasets?page=1 | attribution |
| **PM 2.5** | Global Annual PM2.5 Grids from MODIS, MISR and SeaWiFS AOD, v1 | 2000.01–2015.12 | 0.01°×0.01° | http://fizz.phys.dal.ca/~atmos/martin/?page_id=140 | attribution |
| **dust** | MERRA2 | 2000.01–2015.12 | 0.5°×0.625° | https://disc.gsfc.nasa.gov/datasets?page=1 | attribution |
| **cloud fraction** | Corrected ISCCP and PATMOS-X monthly cloud fraction | 1984.01–2007.12 | 1°×1° | https://rda.ucar.edu/datasets/ds741.5/ | attribution |
| **cloud fraction** | ERA5 | 1979.01–2015.12 | 0.25°×0.25° | https://cds.climate.copernicus.eu | attribution |

| | | | | | |
|---|---|---|---|---|---|
| **total column water vapor** | ERA5 | 1979.01–2015.12 | 0.25°×0.25° | https://cds.climate.copernicus.eu | attribution |
| **TOA ASR** | DEEP-C TOA _ASR v02 | 1985.01–2015.12 | 1°×1° | http://www.met.reading.ac.uk/~sgs02rpa/research/DEEP-C/ | attribution |
| **albedo** | GLASS albedo V05 | 2001.01–2011.12 | 0.05°×0.05° | http://glass-product.bnu.edu.cn | depressing effect |
| **albedo** | CERES EBAF-surface Ed4.0 | 2001.01–2011.12 | 1°×1° | https://ceres.larc.nasa.gov/order_data.php | depressing effect |
| **albedo** | CLARA-SAL | 2001.01–2011.12 | 0.25°×0.25° | https://wui.cmsaf.eu/safira | depressing effect |
| **albedo** | GlobAlbedo | 2001.01–2011.12 | 0.5°×0.5° | http://www.GlobAlbedo.org | depressing effect |
| **surface emissivity** | ASTER_GED v4.1 | 2001.01–2015.12 | 0.05°×0.05° | https://lpdaac.usgs.gov/dataset_discovery/community/community_products_table/ag5kmmoh_v041 | depressing effect |

**Variables: DSR, downward shortwave radiation; DLR, downward longwave radiation; TOA albedo, top of atmosphere albedo; AOD, aerosol optical depth; PM2.5, Particulate matter 2.5; ASR, absorbed solar radiation. Products' name are illustrated in the Section 2.**

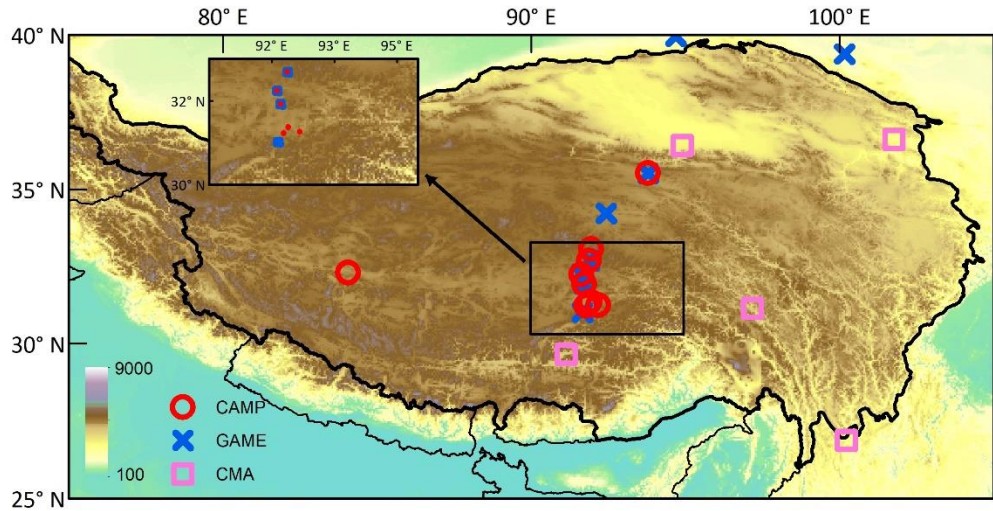


**Figure 1: Site distribution. Observations from three ground networks (Global Energy and Water Exchanges [GEWEX] Asian Monsoon Experiment [GAME], Coordinated Energy and Water Cycle Observation Project [CEOP] Asia-Australia Monsoon Project [CAMP], and China Meteorological Administration (CMA) are from 1960–2005. Vector layer data is free for academic use licensed by Database of Global Administrative Areas (GADM). Elevation data is provided by National Oceanic and Atmospheric**
**Administration (NOAA) (GLOBE).**

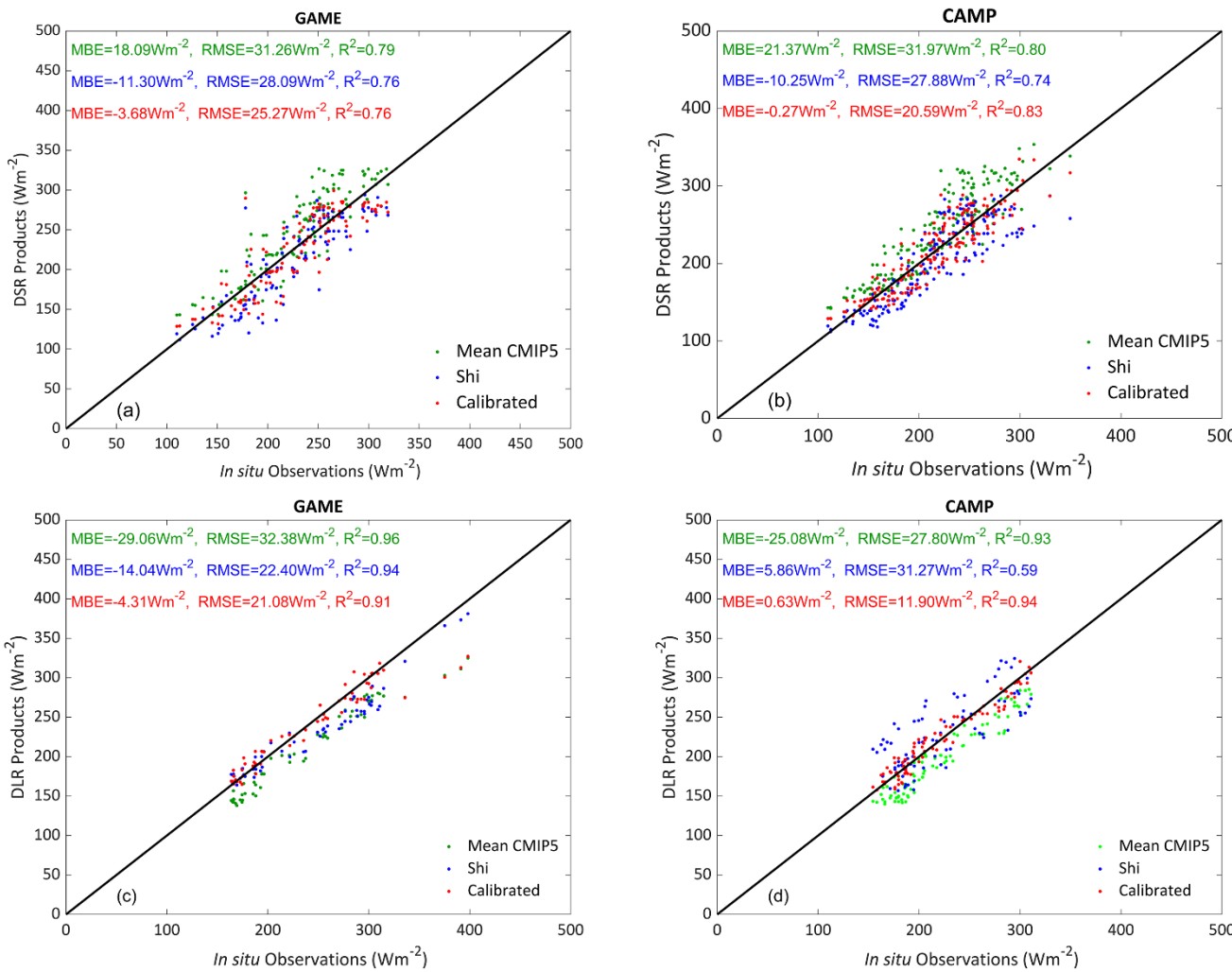

**Figure 2: Scatterplot of site validation results from two ground networks: (a, b) downward shortwave radiation (DSR), (c, d) downward longwave radiation (DLR).**

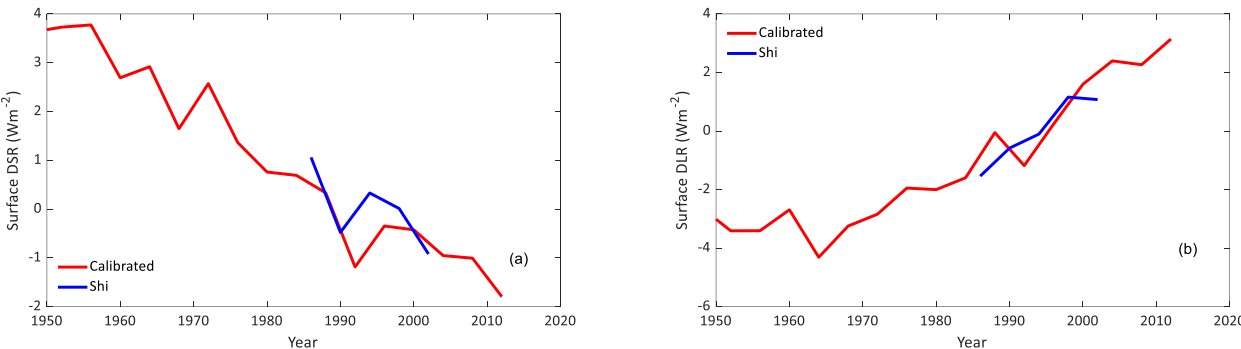

**Figure 3: Trend comparison between calibrated downward radiation datasets and Shi and Liang (2013). Temporal variations over the Tibetan Plateau (TP) in (a) and (b) were averaged by the 5-year moving window in order to remove the impact of annual variability.**

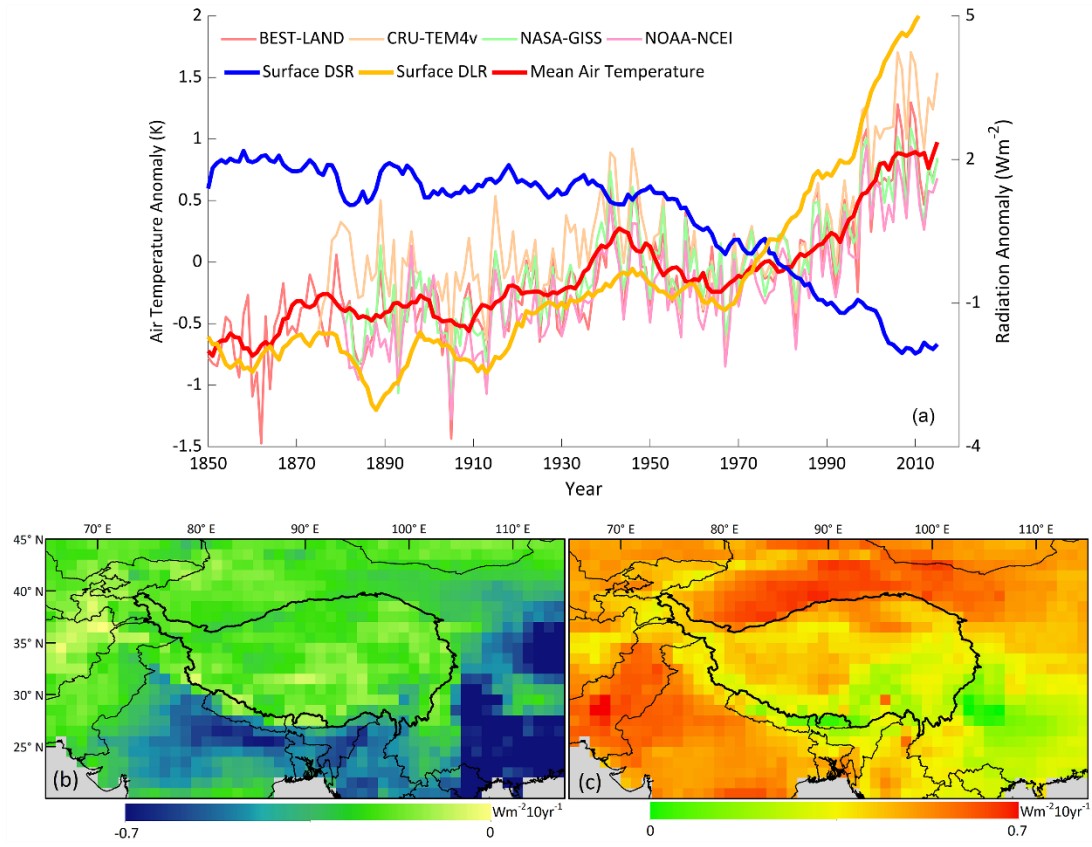

**Figure 4: Spatiotemporal variations in surface downward shortwave radiation (DSR) and downward longwave radiation (DLR) over the Tibetan Plateau (TP) and its neighboring regions from 1850 to 2015 based on calibrated radiation results. Mean air temperature is calculated by the four air temperature datasets. Temporal variations in (a) were averaged by the 10-year moving window in order to remove the impact of annual variability. The trends of DSR (b) and DLR (c) are significant, with p-values < 0.01. Vector layer data is free for academic use licensed by GADM.**

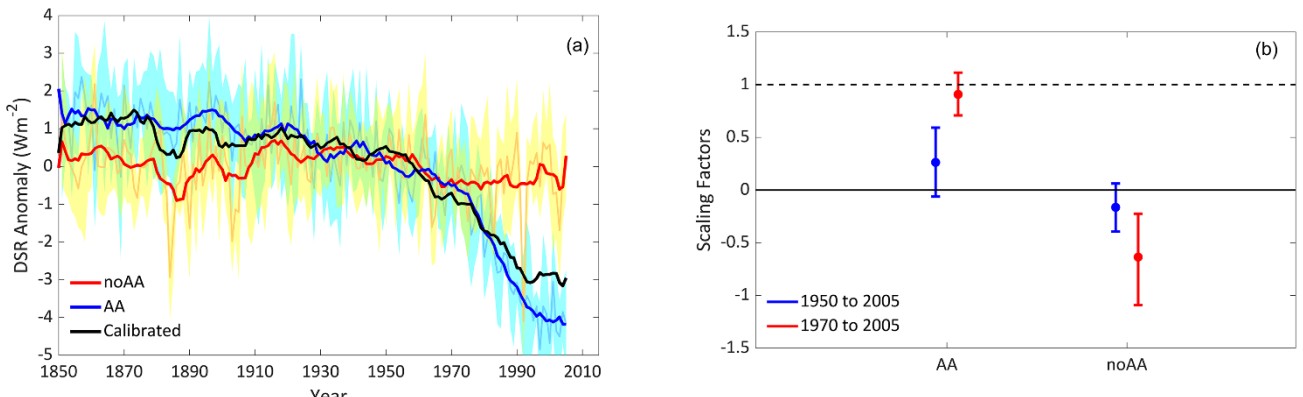

**Figure 5: Temporal variation in detected factors from remote sensing products over the Tibetan Plateau (TP): (a) aerosol optical depth (AOD) and aerosol index, (b) Particulate matter (PM)2.5 and dust, (c) cloud fraction, and (d) top-of-atmosphere (TOA) albedo.**


**Figure 6: (a) Temporal variations of the calibrated, anthropogenic aerosol–driven (AA-driven) and noAA-driven DSR. Temporal variations were averaged by a 10-year moving window to remove the impact of annual variability. The shaded area is the standard deviation of model average. (b) Scaling factors of the AA and noAA forcing simulation on downward shortwave radiation (DSR) based on optimal fingerprinting method. The p value of the impact factor in 1950 – 2005 (1970 - 2005) is 0.22 (0.04).**


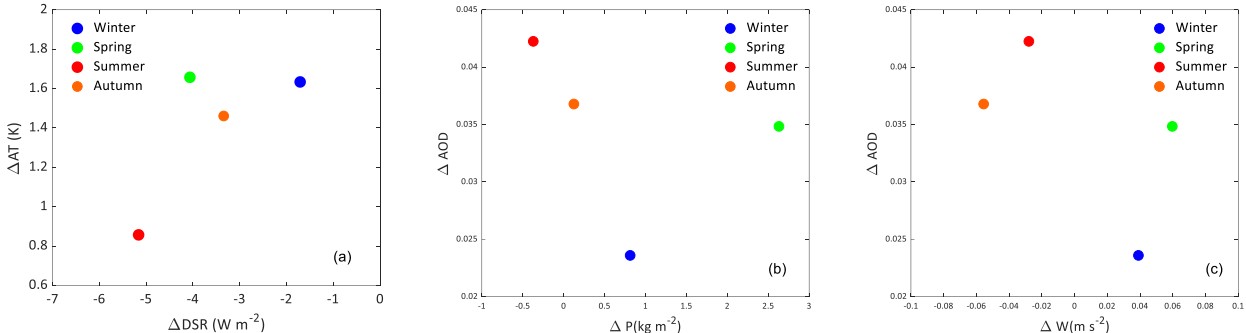

**Figure 7: Relationships between variable changed magnitudes at the seasonal scale from 1850 to 2015: (a) the decrease in downward shortwave radiation (DSR) and increase in air temperature, (b) precipitation change and aerosol optical depth (AOD) increase, (c) wind speed change and AOD increase. Data are from CMIP5 model average.**

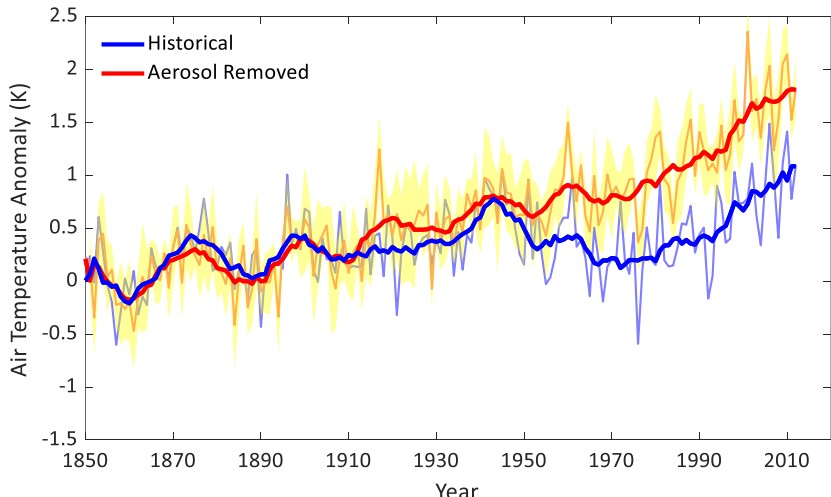

**Figure 8: Temporal annual variation in air temperature, and air temperature with the depressing effect of aerosols removed in the summer season. The shaded area is the standard deviation of model average.**