# Peer review of "Air pollution slows down surface warming over the Tibetan Plateau"

_Atmospheric Chemistry and Physics, 2019_

## Referee Comment (RC1) · Anonymous Referee #1 · 8 Oct 2019

This is a good job. I recommend this article to be published in ACP after addressing the following issues.

Major comments:

1, The Tibetan Plateau (TP) area in your analysis should be defined clearly when you present Fig. 1.

2, The objectives of this paper need to be more clearly stated in the introduction part. Maybe the author needs more references reading.

3, To perform more solid results, some data sets need to be analyzed:

3.1, Please add ERA5 reanalysis data into your analysis. ERA5 can be found at

https://cds.climate.copernicus.eu/#!/search?text=ERA5&type=dataset

3.2, Please use more albedo data such as GlobAlbedo, CLARA-SAL, MODIS......
(see He et al., 2014). He, T., S. Liang, and D.-X. Song (2014), Analysis of
global land surface albedo climatology and spatial-temporal variation during 1981–
2010 from multiple satellite products, J. Geophys. Res. Atmos.,119,10,281–10,298,
doi:10.1002/2014JD021667.

4, The conclusions need to be deepened. Whether other effects also can slow down
surface warming over the TP? Could you conclude that aerosols increase is the major
contribution to surface warming mitigation over the TP? Maybe the author needs to add
more evidence.

Minor comments:

1, Why the first author is not the corresponding author?

2, In the abstract, the time range needs to be specified for the contribution of 48.6%.

3, Please use the orange to replace the yellow color in Fig. 7.

4, Please add more words in the caption of Fig. S3.

5, There is a good review paper including discussions of aerosol effects over the TP
(Qian et al., 2015). Qian, Y., et al.: Light-absorbing particles in snow and ice: Measure-
ment and modeling of climatic and hydrological impact, Adv. Atmos. Sci., 32, 64-91,
2015.

---

## Referee Comment (RC2) · Anonymous Referee #2 · 8 Oct 2019

1. I have pointed out that "No long-term observations of DSR and aerosol data support the long-term variations of DSR and anthropogenic aerosol developed in this study", and the authors chose 5 sites from GEBA to support their main conclusion. But it should be noted that the 5 sites from GEBA is also from the observations of Chinese Meteorological Administration. This contradicts with your statement that "We didn't include ground observations from Chinese Meteorological Administration stations due to data discontinuity and large uncertainty". The obvious low values between 1980 and 1990 is the questionable observations, and this sites can not used to validate the long-term variations of your fused dataset (see Shi et al., 2008).

2. how did you reach that "estimated DSR driven by sunshine duration was not calculated either because the method accuracy may be not high enough to capture the

influence of aerosols at low-level magnitude."? In my opinion, the accuracy of DSR driven by sunshine duration is generally higher than those of satellite-based DSR and CMIP5. At least, the accuracy of DSR driven by sunshine duration is also higher than that fused by yours.

3. As you also known that TP is one of the cleanest areas in the world, and compared to other factors, such as cloud and water vapor, it's effect on the DSR over the TP may be ignorable. Thus, it can not cause the phenomenon of solar dimming over the TP.

4. You did not answer my question fully: "Why did you use the CERES EBAF DSR to calibrate the CMIP5 DSR data since the satellite radiation products generally can not capture the long-term DSR variations. Or you can demonstrate that the CERES EBAF DSR can reflect the long-term variations of DSR?". Even if the CERES EBAF DSR can capture long-term variations of DSR over the other regions, it not necessarily can capture long-term variations of DSR over the TP.

5. Because the 5 sites from the GEBA is measured by the Chinese Meteorological Administration and is the same as the observations of CMA. Thus, the question "The DSR over the Tibetan Plateau is decreased since 1950, which was different with the points derived based on the observations or based on the sunshine based DSR" should be re-answered.

---

## Author Comment (AC1) · 23 Oct 2019

Dear Editor and referees,

We would like to thank you for your valuable comments and suggestions. In this version, we have undertaken several major changes.

First, we clarified the research objectives and expanded the literature review about the aerosol's impacts on temperature based on referee #1's suggestions.

Second, ERA5 and additional surface albedo products have been included in the study according to the suggestion of referee #1.

Third, more discussions were provided about the aerosol depressing effects.

We also replaced GEBA observations by measurements at the CMA sites in the supplementary material, and more explanations were given to address referee #2's concerns about the observation uncertainty, aerosol effects. CERES assessment for capturing the temporal variation has been included in the supplementary suggested from referee #2.

Minor revisions about the grammar and expressions have been done.

The revised manuscript isn't uploaded during the open discussion section based on the journal requirement. Thanks very much for your time and efforts in reviewing the manuscript.

Best,

Aolin Jia and co-authors

**Referee #1:**

This is a good job. I recommend this article to be published in ACP after addressing the following issues.

We appreciate your encouragement and we've given a point-by-point response to the comments as follows.

Major comments:

1, The Tibetan Plateau (TP) area in your analysis should be defined clearly when you present Fig. 1.

Thank you for reminding us. We added the definition of the Tibetan Plateau used in this study at **Line 207-209 (new version)**.

> "The Tibetan Plateau region is defined as the Chinese Qinghai-Tibet Plateau in this paper, covering most of the Tibet Autonomous Region and Qinghai in western China (Wang *et al.*, 2016)."

2, The objectives of this paper need to be more clearly stated in the introduction part. Maybe the author needs more references reading.

Thanks for your suggestions. We clarified our research objectives at **Line 68-70**.

> "In this study, we aim to analyze the long-term spatiotemporal variation of surface radiation over the TP by generating a long-term surface radiation datasets from satellite products and model simulations. Solar dimming is to be attributed by analyzing multiple data sources. The depressing effect of aerosols on climate warming needs to be quantified in the end."

Besides, more literature reviews about the impacts of aerosols on temperatures at different spatial scales were summarized in the introduction (**Line 53-58**) and we point out that currently there is no conclusive answer and is still under discussion.

> "Aerosols have a net cooling effect on the global temperature with higher uncertainty from Intergovernmental Panel on Climate Change (IPCC) report (Stocker et al., 2013), whereas Andreae et al. (2005) has suggested that current aerosol loading may cause a hot future. Even Gettelman et al. (2015) contended that the net effect of aerosols on surface temperature can be neglected, Samset et al. (2018) pointed out that aerosol depressed surface temperature by 0.5-1.1 K globally. By contrast, one recent study (Feng and Zou, 2019) argued that aerosols contributed $+0.005 \pm 0.237$ K on global surface temperature change after 2000. Therefore, the aerosol effect on climate warming is still under discussion."

3, To perform more solid results, some data sets need to be analyzed:

3.1, Please add ERA5 reanalysis data into your analysis. ERA5 can be found at https://cds.climate.copernicus.eu/#!/search?text=ERA5&type=dataset

Thanks for your comment. To take advantage of reanalysis datasets for characterizing atmospheric profiles, we've employed ERA5 into our analysis. First, we replaced the ERA-Interim by ERA5 (the newest version) for detecting the temporal variation of column water vapor over the TP since 1979. The results showed that the variation of ERA5 can match with MODIS atmospheric products very well (SFig. 4, note: we added

new results as SFig 1, so this figure number is changed from 3 to 4. SFig figures in the response file are directly used from Supplementary materials).

[Figure]

SFig. 4. Temporal annual variation of the atmospheric water vapor from MODIS atmospheric products and ERA5. ERA5 shows a considerable turning point in 1998 and the decreasing trend matches with satellite products very well.

The results demonstrated that the column water vapor trend undergoes considerable changes around 1998 and before this year, it had slightly increased and then it decreased significantly. However, solar radiation didn't respond to this variation based on former studies and our results. The overall variation of the column water vapor was not significant in recent 37 years. Therefore, the influence of the column water vapor can be ignored.

We also included the ERA5 in the cloud fraction analysis to prove that cloud coverage over the TP is decreasing (Figure 5c).

[Figure]

Figure 5(c). Temporal variation in detected factors from remote sensing products over the Tibetan Plateau (TP): (c) cloud fraction.

3.2, Please use more albedo data such as GlobAlbedo, CLARA-SAL, MODIS… (see He et al., 2014). He, T., S. Liang, and D.-X. Song (2014), Analysis of global land surface albedo climatology and spatialtemporal variation during 1981– 2010 from multiple satellite products, J. Geophys. Res. atmos.,119,10,281–10,298, doi:10.1002/2014JD021667.

Thanks! We included four surface albedo products (GLASS, CLARA, CERES, GlobAlbedo) to calculate the albedo climatology of the TP. These albedo products cover different satellite observation sources.

First, we generated the monthly climatological albedo of each satellite product, and we computed all standard deviations of any possible three climatological albedo combinations at each pixel. Then for each pixel, we chose the product combination that has the lowest standard deviation and calculated the mean value to represent the ground truth. The final result changed little in the graph especially when it shows the regional averaged depressing impact (Figure 8) because the albedo products have close climatology estimation at mid-latitude as the former study suggested (He *et al.*, 2014). We've added more data description and methodology explanation in the manuscript (**Line 165-183**).

> "According to He *et al.* (2014), the fine-resolution (0.05°) climatological surface albedo products retrieved from satellite observations agree well with each other for all the land cover types in middle to low latitudes. Therefore, we selected four commonly used satellite surface albedo products for calculating the surface albedo climatology over the TP, including the CERES EBAF, the Global LAnd Surface Satellite (GLASS), the Clouds, Albedo, and Radiation-Surface Albedo (CLARA-SAL), and the GlobAlbedo. First, we generated the monthly climatological albedo of each satellite product, and we computed all standard deviations of any possible three product climatology combinations at each pixel. Then for each pixel, we chose the product combination that has the lowest standard deviation and calculated the mean value to represent the ground truth climatology.
>
> …
>
> The CLARA-SAL product is inversed from advanced very high resolution radiometer (AVHRR) observations (Riihelä *et al.*, 2013). Atmospheric correction was done by assuming AOD and ozone is constant. Sensor calibration and orbital drift have been dealt with and the uncertainty of monthly albedo estimation is about 11%. The GlobAlbedo product uses an optimal estimation approach European satellites, including Advanced Along-Track Scanning Radiometer (AATSR), SPOT4-VEGETATION, SPOT5-VEGETATION2, and Medium-Resolution Imaging Spectrometer (MERIS) (Lewis *et al.*, 2013). MODIS surface anisotropy information was used for gap-filling. More detailed algorithm introductions and comparison can be found in (He *et al.*, 2014)."

4, the conclusions need to be deepened. Whether other effects also can slow down surface warming over the TP? Could you conclude that aerosols increase is the major contribution to surface warming mitigation over the TP? Maybe the author needs to add more evidence.

Thanks for your suggestion. We've added more discussions about the depressing effect of aerosols and other factors in terms of water vapor (**Line 431-539**).

> "The attribution of solar dimming over the TP and corresponding aerosol effect quantification revealed that anthropogenic aerosols dominate the solar radiation decrease and depress the climate warming in recent decades. Aerosols are cloud condensation nuclei (CCN) and more CCN may depress the cloud formation and precipitation. Thus future studies need to analyze the indirect effect of aerosol loading (Qian *et al.*, 2015) over there. However, it should be noticed that we don't conclude that TP undergoes warming mitigation. In fact, TP has a rapid warming rate than global warming (Yao *et al.*, 2018) and other varying factors also affect the warming

rate, in terms of the water vapor variation around 1998 (Supplementary Fig. S3). Water vapor is a weak DSR-absorbing factor but a major greenhouse gas emitting downward longwave radiation, so its decrease might slow down the local warming rate. However, the impact of water vapor variation after 1998 is at an annual scale that cannot match the analysis in this study, thus more further researches may focus on it."

We would like to point out that the TP didn't undergo temperature mitigation. For water vapors, we tried to conduct depressing analysis but there are some limitations. Reanalysis datasets start from 1980 while our depressing analysis focused on decade scales (Figure 8). Besides, CMIP5 didn't release any column water vapor variables or provide some HistoricalMisc experiments designed for water vapors' influences, which means it's hard to conduct related attribution analysis and depressing quantification based on CMIP5 experiments at decadal scales. Therefore, in this study, we only focus on aerosol impacts on climate warming.

We mentioned in the discussion that varying column water vapor trends in 1998 (SFig. 4) could cause some impacts on local warming because it is a weak shortwave absorber but an important greenhouse gas emitting longwave radiation. The decreasing water vapor may depress the local warming and the follow-up researches can work on it at annual scale. In this study, we mainly focus on the impact of aerosols on the long-term temporal scale.

Minor comments:

1, Why the first author is not the corresponding author?

Prof. Liang is the advisor of the first author Aolin Jia who is currently a Ph.D student.

2, In the abstract, the time range needs to be specified for the contribution of 48.6%.

Corrected. Thanks!

3, Please use the orange to replace the yellow color in Fig. 7.

Corrected. Thanks!

4, Please add more words in the caption of Fig. S3.

Corrected. Thanks!

5, There is a good review paper including discussions of aerosol effects over the TP (Qian et al., 2015). Qian, Y., et al.: Light-absorbing particles in snow and ice: Measurement and modeling of climatic and hydrological impact, Adv. Atmos. Sci., 32, 64-91, 2015.

We included it, thanks for your help!

**References**

Andreae, M. O., C. D. Jones, and P. M. Cox (2005), Strong present-day aerosol cooling implies a hot future, Nature, 435(7046), 1187.

Feng, H., and B. Zou (2019), Satellite-based estimation of the aerosol forcing contribution to the global land surface temperature in the recent decade, Remote Sensing of Environment, 232, 111299.

Gettelman, A., D. Shindell, and J.-F. Lamarque (2015), Impact of aerosol radiative effects on 2000–2010 surface temperatures, Climate dynamics, 45(7-8), 2165-2179.

He, T., S. Liang, and D. X. Song (2014), Analysis of global land surface albedo climatology and spatial‐temporal variation during 1981–2010 from multiple satellite products, Journal of Geophysical Research: Atmospheres, 119(17), 10,281-210,298.

Lewis, P., C. Brockmann, and J. Muller (2013), GlobAlbedo: Algorithm theoretical basis document V4-12, in Technical Report, edited.

Qian, Y., T. J. Yasunari, S. J. Doherty, M. G. Flanner, W. K. Lau, J. Ming, H. Wang, M. Wang, S. G. Warren, and R. Zhang (2015), Light-absorbing particles in snow and ice: Measurement and modeling of climatic and hydrological impact, Advances in Atmospheric Sciences, 32(1), 64-91.

Riihelä, A., T. Manninen, V. Laine, K. Andersson, and F. Kaspar (2013), CLARA-SAL: a global 28 yr timeseries of Earth's black-sky surface albedo, Atmospheric Chemistry and Physics, 13(7), 3743-3762.

Samset, B., M. Sand, C. Smith, S. Bauer, P. Forster, J. Fuglestvedt, S. Osprey, and C. F. Schleussner (2018), Climate impacts from a removal of anthropogenic aerosol emissions, Geophysical Research Letters, 45(2), 1020-1029.

Stocker, T. F., D. Qin, G.-K. Plattner, M. Tignor, S. K. Allen, J. Boschung, A. Nauels, Y. Xia, V. Bex, and P. M. Midgley (2013), Climate change 2013: The physical science basis, edited, Cambridge University Press Cambridge.

Wang, Z., Z. Li, M. Xu, and G. Yu (2016), River morphodynamics and stream ecology of the Qinghai-Tibet Plateau, CRC Press.

Yao, T., Y. Xue, D. Chen, F. Chen, L. Thompson, P. Cui, T. Koike, W. K.-M. Lau, D. Lettenmaier, and V. Mosbrugger (2018), Recent Third Pole's rapid warming accompanies cryospheric melt and water cycle intensification and interactions between monsoon and environment: multi-disciplinary approach with observation, modeling and analysis, Bulletin of the American Meteorological Society(2018).

---

## Author Comment (AC2) · 23 Oct 2019

Dear Editor and referees,

We would like to thank you for your valuable comments and suggestions. In this version, we have undertaken several major changes.

First, we clarified the research objectives and expanded the literature review about the aerosol's impacts on temperature based on referee #1's suggestions.

Second, ERA5 and additional surface albedo products have been included in the study according to the suggestion of referee #1.

Third, more discussions were provided about the aerosol depressing effects.

We also replaced GEBA observations by measurements at the CMA sites in the supplementary material, and more explanations were given to address referee #2's concerns about the observation uncertainty, aerosol effects. CERES assessment for capturing the temporal variation has been included in the supplementary suggested from referee #2.

Minor revisions about the grammar and expressions have been done.

The revised manuscript isn't uploaded during the open discussion section based on the journal requirement. Thanks very much for your time and efforts in reviewing the manuscript.

Best,

Aolin Jia and co-authors

**Referee #2:**

1. I have pointed out that "No long-term observations of DSR and aerosol data support the long-term variations of DSR and anthropogenic aerosol developed in this study", and the authors chose 5 sites from GEBA to support their main conclusion. But it should be noted that the 5 sites from GEBA is also from the observations of Chinese Meteorological Administration. This contradicts with your statement that "We didn't include ground observations from Chinese Meteorological Administration stations due to data discontinuity and large uncertainty". The obvious low values between 1980 and 1990 is the questionable observations, and this sites cannot be used to validate the long-term variations of your fused dataset (see Shi et al., 2008).

Thanks for the valuable comment. We've corrected this mistake. In the revised version, we employed CMA rather than GEBA data. Only observations before 1980 are used in order to avoid the data discontinuity issue after 1980. The revised results show that surface DSR observations can reflect TP dimming since 1958 with large uncertainty [SFig. 2(a), SFig mark means the figure is shown in the supplementary material and directly used here.]

[Figure]

SFig. 2(a). Surface DSR temporal variation of (a) 5 CMA sites mean, (b-f) individual sites. Temporal variations were averaged by the 5-year moving window in order to remove the impact of annual variability.

We still use surface radiation measurement as a reference. 130 CMA radiation sites over China were collected and 12 sites are located in the TP. For detecting the long-term DSR variation, the sites starting to operate after 1970 were not used, so 7 sites left (Figure r1, following figures marked by 'r' are only shown in the response file).

[Figure]

Figure r1. 7 CMA sites distribution.

We drew the averaged DSR temporal variation at 7 sites, and corresponding site numbers at each year is shown by red bars.

[Figure]

Figure r2. DSR temporal variation at (a) 7 sites, (b) 6 sites without site 56029.

In Figure r2, the dimming time of 7 sites started in 1967, which is different from our study, and the unstable annual anomalies in 1958-1960 and 1968-1970 are mainly caused by missing measurements in some sites in these years. However, sites 56029 and 55299 had continuously missing measurements for more than 5 years. Therefore, considering the data continuity and location sampling (site 56029 is near site 56137, and site 55299 is near site 55591 compared with other sites), we abandoned these two sites in SFig. 2, and the left 5 sites are scattered in TP.

In the SFig. 2, the dimming time started in 1958, and only site 55591 has a different starting time. Sites [56651 and 52866, SFig. 2(c, f)] located in eastern region show clear DSR decreases from 1958, and the other 2 sites have an overall slight decrease with oscillation from 1958. It is consistent with our result (Figure 4a) that TP dimming is more significant in the southeastern region.

We also checked the dimming time change in site averaged results. We found that once site 56029 was removed in the analysis, the starting time would be changed to 1958 [Figure r2 (b)]. It illustrated that the site number and location did considerably affect the starting time. Our data covered the whole TP and caught the solar decrease especially at southeast TP.

In all, both site observations and our results can prove that the TP has undergone dimming since the 1950s. The large uncertainty of site observations and larger dimming trend may be caused by measurement drifts explained by He *et al.* (2018) who used sunshine duration derived DSR showing a smaller dimming magnitude compared with observed DSR at global scale.

2. how did you reach that "estimated DSR driven by sunshine duration was not calculated either because the method accuracy may be not high enough to capture the influence of aerosols at low-level magnitude."? In my opinion, the accuracy of DSR driven by sunshine duration is generally higher than those of satellite-based DSR and CMIP5. At least, the accuracy of DSR driven by sunshine duration is also higher than that fused by yours.

We speculated that Sunshine Duration (SunDu) derived DSR in TP cannot capture the trend at the decadal scale and SunDu may not represent DSR to show TP dimming especially for the early period at the TP based on the results in He *et al.* (2018). In their study, He et al. estimated DSR from SunDu from globally distributed site observation pairs based on a widely used method (Kun Yang *et al.*, 2006), and observed DSR is considered as reference and the estimation accuracy is satisfactory at the global scale.

However, we found that the SunDu derived DSR has an opposite trend with observed DSR in TP [Figure r3, also Figure 3 in (He *et al.*, 2018)].

[Figure]

Figure r3. Maps of the decadal trends (units: W/m2 per decade) in 2.5° × 2.5° grids of sunshine duration (SunDu)-derived Rs (a and d), the observed Rs (b and e), and differences between the two data sets (c and f) over China and Europe during two periods of dimming and brightening. "Dimming" denotes the periods of 1959–1989 in China and 1961–1980 in Europe. "Brightening" denotes the periods of 1994–2010 in China and 1980–2009 in Europe.

In Figure r3, the dimming trend from SunDu-derived DSR matched with observed DSR except over the TP region. The paper didn't provide more explanations about the mismatch. However, when they applied this method in more than 2000 sites over china, we found that their SunDu-derived DSR over the TP has no dimming at all time periods (Figure r4, also Figure 4 in (He *et al.*, 2018)).

[Figure]

Figure r4. Maps of the decadal trends (units: W/m2 per decade) of all reliable SunDu-derived Rs stations over China, Europe, and the United States in 2.5° × 2.5° grids during three periods. "dimming" denotes the periods of 1959–1989, 1950–1980, and 1952–1980 in China, Europe, and the United States, respectively. "Brightening" denotes the periods of 1994–2010 in China and 1980–2009 in Europe.

They didn't focus on TP so there is no specific explanation of it, but the result is contradictory with our result and all former studies based on direct observations, model simulations, reanalysis, and satellite

observations (Kuang and Jiao, 2016; Shi and Liang, 2013; K. Yang *et al.*, 2012; K. Yang *et al.*, 2014; You *et al.*, 2010).

We also contacted with the co-author Martin Wild who is in charge of GEBA network and he also expects that the sign of DSR derived from SunDu is same as the DSR observations.

Dear Aolin Jia,

Thanks for you mail. The Chinese data in GEBA have not been changed with respect to the CMA original data.
There are problems in the Chinese radiation data quality as you are sure aware, and as documented in many papers.
I had a visitor from CMA (Yang Su) visiting me for a year and working with me on the improvement of the quality of the dataset. I attach 2 related papers recently published in J. Climate.
Unfortunely those data are not public due to the Chinese data policy.
As for the Sunshine duration trends, I also expect them to be of the same sign as the radiation data. You may contact the Beijing group for more details on their analysis.

Kind regards
Martin
* * *
Prof. Martin Wild
Institute for Atmospheric and Climate Science
ETH Zurich
Universitaetsstr. 16
CH-8092 Zurich (Switzerland)

Figure r5. Email from Martin Wild

Besides, we discussed this issue with the first author of the paper, who provided some valuable details about the estimated DSR over the TP. They explained that studying different time periods may result in different trends, it's true but unfortunately it cannot explain that why the overall trend at two time period is brightening especially at southeastern TP [Figure r6 (b), site 56651 is at the southeastern TP while the trend is overall negative]. The sites [Figure r6 (a)] they provided showed that these sites have dimming trend that matched former results, while the whole trend shown in Figure r4 is still brightening at dimming period (1952 - 1989), which is different from the DSR observations [SFig. 2 (a)]. We infer that even for the SunDu sites, the dimming time varied at different locations that matched what we found using DSR observations.

[Figure]

Figure r6. (a) Site samples the author provided for us; (b) DSR temporal variation of CMA 56651. Temporal variations were averaged by the 5-year moving window in order to remove the impact of annual variability.

Therefore, we speculated that SunDu-derived DSR couldn't be able to capture the observed DSR temporal variation in TP and SunDu may not represent DSR to show the dimming at the TP. According to Manara *et al.* (2017), SunDu has a different sensitivity to atmospheric turbidity changes that is estimated by aerosol optical depth (AOD). SunDu may lose its representability at low AOD level. We infer that this is the reason why this method didn't capture the dimming trend in the TP and SunDu may not represent DSR in the TP at low AOD level.

Additionally, the accuracy (standard deviation of bias, STD) of SunDu-derived DSR over China is about 19.32 Wm$^{-2}$ (He *et al.*, 2018), and our validation showed that the standard deviation of the calibrated data bias is 20.64 Wm$^{-2}$. Considering that the validation of gridded data has scale mismatch effect while their validation results are observation pairs, we think our result is comparable to theirs. Besides, their validation sampling is over China while our validation samples are only limited in TP, where DSR is large and the bias and STD could be larger, let alone the measurement environment in TP is not as good as other regions and might introduce large uncertainty.

More discussions of physical relationship between DSR and SunDu and the estimation algorithm suitability are beyond the scope of this study, therefore, we didn't include more experiments assessing the estimation algorithm and directly used DSR observations as the references.

3. As you also known that TP is one of the cleanest areas in the world, and compared to other factors, such as cloud and water vapor, it's effect on the DSR over the TP may be ignorable. Thus, it can not cause the phenomenon of solar dimming over the TP.

Thanks! When we estimate instantaneous DSR at all-sky conditions, it's reasonable to ignore the influence of AOD because its influence is small compared to the DSR absolute value. However, when we analyze the impact at the decadal scale, any contributing factor that has a directional decrease or increase trend will affect the DSR trend accordingly. We also calculated the radiative effect of aerosols in Figure 6 (a), ~5 Wm$^{-2}$ difference of decadal variation between the clean and aerosol case simulations cannot be ignored.

Besides, we also calculated the increased aerosol radiative forcing caused by AOD increase since 1998 based on K. Yang *et al.* (2012). The increased radiative forcing is about 1.97 Wm$^{-2}$, which can also prove that it is not ignorable.

As we explained in the last reply, it's true that TP is one of the cleanest areas in the world and the corresponding aerosol climatology is low. However, when we talk about solar dimming over the TP, we mainly focus on the DSR decreasing phenomenon over there, which is characterized by the variation of DSR decadal anomalies rather than the absolute magnitude. Besides, it's necessary to point out that when aerosol loadings in the atmosphere are at a low magnitude, direct radiative effects (scattering and absorption effect) play a dominant role in the interaction between aerosols and the atmosphere (Li *et al.*, 2017). Therefore, even TP has a clean condition, it is easily affected by aerosols increase.

4. You did not answer my question fully: "Why did you use the CERES EBAF DSR to calibrate the CMIP5 DSR data since the satellite radiation products generally can not capture the long-term DSR variations. Or you can demonstrate that the CERES EBAF DSR can reflect the long-term variations of DSR?". Even if the CERES EBAF DSR can capture long-term variations of DSR over the other regions, it not necessarily can capture long-term variations of DSR over the TP.

Thanks for your comment. We understand your concern.

First we've proved in the previous reply (Figure r7) that CERSE EBAF 4.0 can capture the absolute value variation over the CAMP network in the TP even there is a systematic bias at some sites.

[Figure]

Figure r7. (a) Taylor diagram of solar validation of CERES EBAF (**black dot C**) and 18 CMIP5 models (**grey dots**). (b) Monthly variation of CERES EBAF (**blue line**) and site observations (**red line**). Only sites that were run more than 2 years long were shown here.

Then we used 11 CMA sites located in TP (deleted one that missed continuous measurement for 3 years) to prove that CERES EBAF 4.0 DSR can capture the overall temporal annual anomaly variation observed by CMA since 2001 over the TP (SFig. 1). Therefore, we can choose CERES as the reference at each pixel to calibrate the model simulation results. We added the CERES analysis into the supplementary as SFig. 1. Thanks for reminding us.

[Figure]

[Figure]

Fig. S1: Surface DSR temporal variation of CERES and all CMA radiation sites at TP (a) 11 CMA sites mean, (b-l) individual sites, and (m) 11 sites distribution

Thanks very much for providing this valuable suggestion.

Base on the analysis in Q2, we suggested that the different start time of TP dimming from the previous study based on SunDu and our result is caused limited representability of SunDu to DSR at the TP. Therefore, we still selected DSR measurement as ground reference data. By analyzing DSR measurements, we concluded that sites at different locations show various dimming start time. Our data can cover the whole TP area especially the southeastern TP. Thus they may have a different start time.

**References**

He, Y., K. Wang, C. Zhou, and M. Wild (2018), A revisit of global dimming and brightening based on the sunshine duration, Geophysical Research Letters, 45(9), 4281-4289.

Kuang, X. X., and J. J. Jiao (2016), Review on climate change on the Tibetan Plateau during the last half century, J Geophys Res-Atmos, 121(8), 3979-4007, doi: 10.1002/2015jd024728.

Li, Z., D. Rosenfeld, and J. Fan (2017), Aerosols and their impact on radiation, clouds, precipitation, and severe weather events, in Oxford Research Encyclopedia of Environmental Science, edited.

Manara, V., M. Brunetti, M. Maugeri, A. Sanchez‐Lorenzo, and M. Wild (2017), Sunshine duration and global radiation trends in Italy (1959–2013): To what extent do they agree?, Journal of Geophysical Research: Atmospheres, 122(8), 4312-4331.

Shi, Q. Q., and S. L. Liang (2013), Characterizing the surface radiation budget over the Tibetan Plateau with ground-measured, reanalysis, and remote sensing data sets: 2. Spatiotemporal analysis, J Geophys Res-Atmos, 118(16), 8921-8934, doi: 10.1002/jgrd.50719.

Yang, K., T. Koike, and B. Ye (2006), Improving estimation of hourly, daily, and monthly solar radiation by importing global data sets, Agricultural and Forest Meteorology, 137(1-2), 43-55.

Yang, K., B. H. Ding, J. Qin, W. J. Tang, N. Lu, and C. G. Lin (2012), Can aerosol loading explain the solar dimming over the Tibetan Plateau?, Geophysical Research Letters, 39, doi: Artn L2071010.1029/2012gl053733.

Yang, K., H. Wu, J. Qin, C. G. Lin, W. J. Tang, and Y. Y. Chen (2014), Recent climate changes over the Tibetan Plateau and their impacts on energy and water cycle: A review, Global Planet Change, 112, 79-91, doi: 10.1016/j.gloplacha.2013.12.001.

You, Q. L., S. C. Kang, W. A. Flugel, A. Sanchez-Lorenzo, Y. P. Yan, J. Huang, and J. Martin-Vide (2010), From brightening to dimming in sunshine duration over the eastern and central Tibetan Plateau (1961-2005), Theoretical and Applied Climatology, 101(3-4), 445-457, doi: 10.1007/s00704-009-0231-9.

---

## Short Comment (SC1) · 4 Nov 2019

[supplement omitted: unrelated document]

---

## Short Comment (SC2) · 4 Nov 2019

[supplement omitted: unrelated document]

---

## Author Comment (AC3) · 11 Nov 2019

Dear Editor and the reader,

We would like to thank you for your positive comments and detailed suggestions. In this manuscript, we have undertaken several changes.

First, we provided the significance test and p-values for all regression analyses.

Second, the assessment analysis and discussion about satellite climatology have been added.

Third, the methodology and data description have been revised.

After making substantial improvements, we have addressed your comments in this

revised manuscript. Please find the specific response to comments in the following context. The revised parts in the manuscript are marked.

Thanks very much for your time and efforts in reviewing the manuscript.

Best, Aolin Jia and co-authors

Please also note the supplement to this comment:
https://www.atmos-chem-phys-discuss.net/acp-2019-553/acp-2019-553-AC3-supplement.pdf

**Supplement:**

Dear Editor and the reader,

We would like to thank you for your positive comments and detailed suggestions. In this manuscript, we have undertaken several changes.

First, we provided the significance test and p-values for all regression analyses.

Second, the assessment analysis and discussion about satellite climatology have been added.

Third, the methodology and data description have been revised.

After making substantial improvements, we have addressed your comments in this revised manuscript. Please find the specific response to comments in the following context. The revised parts in the manuscript are marked.

Thanks very much for your time and efforts in reviewing the manuscript.

Best,

Aolin Jia and co-authors

The Tibetan Plateau is a region which undergoes significant climate change. Air temperatures have increased with 1.39 K since 1850 while the amount of incident solar radiation decreased. The consequences of this solar dimming phenomenon on surface warming are still unclear. Previous research shows contradictory conclusions regarding the proper attribution of solar dimming. Therefore, the roles of clouds and aerosols will be investigated in this study to provide more clarity regarding the causes and impacts of solar dimming.

The paper is well written and the different sub-sections improve the readability and enable the reader to search for specific sections. I feel confident about the data analysis and interpretation done by the authors. However, there are some important remarks regarding certain assumptions, significance of results and data visualisation. I would strongly recommend considering and including these remarks in the manuscript before publication. I will come back to these remarks in more depth in the remainder of this review. Firstly, I want to emphasise what I thought to be very good and interesting about this research. To start with the introduction which describes in a clear and convincing way why this research is relevant. The current controversy regarding the proper attribution of solar dimming is a driving force for this research to introduce new knowledge and provide a conclusive answer. In order to generate this new knowledge, multiple high-quality data sources have been used: model simulations, remote sensing products and ground measurements. The methods applied seem quite advanced and are well-documented in previous literature which makes the methods trustworthy because it can be checked and compared with other research. Especially the improved accuracy of the generated downward surface radiation datasets by applying the NNLS method is a very strong aspect of this research. The solar dimming phenomenon has a large effect on local but also on global climate change. It turns out that humans are largely responsible for the increase of air pollution which turns out to be the main driving factor of solar dimming. The role of human activities in remote areas is discussed and emphasises the societal relevance of the topic.

We greatly appreciate your positive comments.

**Major argument 1:**

The method which is used for the attribution analysis of solar dimming is the optimal fingerprinting method. It's is based on a linear relationship between driving variables and a responding variable, in this case downward shortwave radiation (DSR). When the scaling factor is larger than zero at a certain significance level, the variable has a positive contribution towards the responding variable. My concern regarding this method is that no value of the significance level is given in the manuscript. The results of the attribution analysis indicate that anthropogenic aerosols (AA) are the main cause of solar dimming. However, it's not clearly described or listed if other variables were tested with optimal fingerprinting method besides the noAA simulation and if there were variables which didn't reach the required significance level and are consequently left out of the analysis. The CMIP5 simulations with and without AAs have uncertainties which are indicated by the shaded area in figure 6a. Zhou et al. (2018) calculated the 5%-95% confidence intervals using Monte Carlo simulations. Do these shaded areas and errors bars represent the same confidence intervals and are they calculated in a similar manner? It is stated that the overall variation is of significance tested but the outcomes of these statistical tests or thresholds (p-values, r-values, etc.) are not included. The time over which the method is applied is divided in two periods: 1950-2005 and 1970-2005. Is the selection of these periods linked with the respective increase and decrease of downward longwave and shortwave radiation? Do you believe that two periods are enough to describe

the trend in the data? Yao et al. (2018) described for example that the heating of the Tibetan Plateau began in the 1960s but reached the highest levels in the last 30 years which indicates that significant changes in the climate have occurred within the selected periods.

The results show that the scaling factor is positive for the AA simulation and negative for the noAA simulation, which supports the conclusion that AAs are the main driver of solar dimming. Especially the scaling factor for the AA simulation for 1970-2005 seems convincing with small error bars and a mean value close to 1. If other variables would have been included in the analysis, the scaling factors could be compared with the scaling factor of the AA simulation. This would show the relative contribution of other factors and possibly strengthen the assumption that AAs are indeed the main driving factor. It can be observed that the scaling factors become more positive and more negative for the shorter time period. The error for noAA (1970-2005) is quite substantial in my opinion because the total length of the error bars covers approximately 1/3 of the length of the y-axis of the graph. The negative scaling factor for noAA is attributed to the decrease in cloud cover. The evidence for this statement is obtained from figure 5c where the temporal variation in cloud cover from three satellite data sources is shown. However, the satellite data only covers the period 1980-2005/2015. Thus, from the period 1950-1980 there is no data available to support this conclusion. In addition, the trend of the ISCCP data shows a slight increase of cloud cover which doesn't support the statement that the cloud fraction decreased over time.

I would recommend providing the value(s) of the significance level in the methodology section and indicate if certain variables were left out of the analysis. Could these variables be included when the significance level would change and would this make a difference for the outcomes of the analysis in your opinion? The results of the analysis would be more robust if the values of statistical tests and thresholds are included with the results and figures in the manuscript. Currently I have to believe that the variation is of significance tested without this statement being supported by numbers. Could you elaborate a bit more the selection of the two different time periods in the methodology section, why did you choose for these periods? Am I correctly assuming that it's related to the turning-point of the increase of longwave radiation and the decrease of shortwave radiation? The negative scaling factor of the noAA simulation, with the largest uncertainty, is completely attributed to the decrease in cloud cover which is supported by 2 out of 3 data sources whereas the third data source indicates a slight increase in cloud cover. What is your opinion on the controversy regarding these results and do you have possible suggestions for other factors besides cloud cover which could play a role? Perhaps it would be nice to add a paragraph of discussion concerning the remarks related to this argument in the manuscript.

Thank you for suggesting to include the significance level and we've added the explanation into the methodology (2.2.2). The shaded area in all figures is the standard deviation of model average at each year and we added the explanation in Figure 6 caption. For the (b), we also used Monte Carlo simulations to quantify the uncertainty at 5% - 95% significance level. The p-value of the impact factor in 1950 – 2005 (1970 - 2005) is 0.216 (0.042). The impact factor in 1970 - 2005 passed the significance test. We also added significance statistics (p-value) in other figures.

The introduction of the optimal fingerprinting method has been revised (2.2.2). $X_i$ in the formula are the DSR simulation results from averages of aerosol-driven experiment ensembles and non-aerosol-driven experiment ensembles in this study. It's not reasonable by directly including natural factors into the formula because a simple coefficient cannot represent the relationship between driving factors and DSR. Therefore, the historical DSR is the weighted average of DSR simulation at different forcing cases. The

HistoricalMsic experiments didn't release DSR simulation results for all atmospheric factors (e.g. water vapor, cloud cover) and mainly focused on anthropogenic forcings (e.g. AA, Ozone, and $CO_2$, ...). Therefore, for the solar dimming attribution, we only used AA and noAA HistoricalMisc experiment in the study and assumed that noAA experiment can represent cloud/water vapor impacts on surface downward shortwave radiation.

We applied the optimal fingerprint analysis for two time periods because we found that the impact of AA after 1950s is not large enough or significant (Figure 6b): the impact factor is small and p-value is larger than 0.05. We think it is because of the time period between 1950-1970 when AA, noAA, and historical simulations all have a similar slowly decreasing trend (Figure 6a). Then we ignored this time period (1950 - 1970) and focused on the time span after 1970 to check the corresponding impact and significance because after this year when noAA is pretty much stable while AA and historical records are decreasing considerably (Figure 6a). We didn't include the time period since 1980 and afterward because 1) the speed wasn't accelerated and 2) the time span is half of the former ones that the statistical amount is not comparable. When the statistical number is small, we found the statistics became unstable and easily affected by annual anomalies. Therefore, after 1980, we prefer to use satellite products and reanalysis datasets to demonstrate our analysis.

Additionally, we inferred that the negative impact of noAA is mainly affected by decreasing cloud coverage with a larger uncertainty bar since 1970. As a matter of fact, DSR was driven by more forcings in noAA than AA experiments, introducing more uncertainties among model simulations after 1970. Therefore, it is possible that noAA impact factor shows a larger uncertainty bar. We inferred that cloud coverage dominated the negative natural impact because water vapor has been quite stable since 1980s (Supplementary Fig. 4) that had little impact on the dimming. More explanations about this concern were provided in the corresponding content (Line 401 - 407). Your concern is valuable to this study.

As for the cloud cover variation, we followed referee#1's suggestion and included ERA5 as a long-term dataset. It matches our results:

[Figure]

Figure 5(c). Temporal variation in detected factors from remote sensing products over the Tibetan Plateau (TP): (c) cloud fraction.

The trend (1984 - 2015) of ISCCP is 0.068% per decade but the p-value is 0.80; PATMOS-X is -0.754% per decade; ERA5 is -0.024% per decade but the p-value is 0.62; and CERES is -0.843% per decade (2001-2015) as a reference. 3 of 4 products meet our assumption (except ISCCP) while ISCCP can match with CERES

well and the overall trend of ISCCP and CERES is negative. Besides, the trend value is affected by the annual anomaly and the beginning year. They all have a negative slope if we choose the time span since 1985 and all significantly decreased while starting since 1989. At least all the long-term cloud products were included for proving that the cloud coverage is not the dimming driver.

In fact, the cloud cover decrease over the TP is not a new argument and former studies also found cloud coverage decrease at site scale, which is consistent with the satellite observations (Kuang and Jiao, 2016; Yang *et al.*, 2012). Therefore, there are some site observations supporting the cloud coverage decrease before 1980.

We didn't aim to duplicate the site analysis, thus we calculated the temporal variation of the regional averaged cloud average by using revised long term satellite products. It's the first time people use revised cloud products to analyze the cloud change over the TP. ISCCP wasn't excluded from the analysis because we need to demonstrate the uncertainties among long-term datasets and we don't want to only keep the evidence that strongly supports our results. We added more discussions in the manuscript. Thanks for your suggestions.

Major argument 2:

Shortwave and longwave radiative effects are separated in order to quantify the depressing effect of aerosols on surface warming. It is assumed that the change in air temperature is dominated by the change in surface skin temperature interacting with the air temperature through radiative and thermal processes and the change in atmospheric circulation. Consequently, the variable f is calculated which represents the sensitivity of air temperature to 1 W/m2 radiative forcing. For this analysis I'm wondering whether it's valid to employ values of $\alpha$, $\varepsilon y$ and S which are calculated by taking the mean values of satellite products for several years. Is there a substantial variation between different products and how large or small is the error estimate of this mean value? From the introduction and other studies, it becomes clear that this region undergoes significant climate change which is supported for example by the analysis done by Yao et al. (2018) regarding oxygen isotopes in ice cores collected at glaciers at various locations. The Tibetan Plateau contains large amounts of snow and ice and is called the Third Pole for a reason. Warming and consequent melting of snow and ice could substantially change the albedo. The positive snow-albedo feedback could accelerate the change in albedo and warming over the Tibetan Plateau (Zhang et al. 2003). In addition, other studies indicate that black carbon (BC) and dust are responsible for about a 20% reduction of the albedo (Schmale et al. 2017). However, the results in this study show that the amount of dust decreased over time. Is the amount of BC somehow related to the amount of PM2.5 and could this be responsible for the decrease in albedo besides the decrease due to snowmelt? My main concern regarding this method is whether it's valid to assume that a mean value can represent the rapid changes caused by a positive feedback mechanism in combination with other factors like dust and BC. Additionally, it's not clearly stated over how many years this average is taken, if multiple averages were used for different time periods and which satellite products were used.

The results show that the aerosol radiative forcing has been increasing by 8.08 W/m2 between the first and last 30 years of climatology. However, the supplementary figure S5 shows a negative forcing anomaly which implies a decrease of the radiative forcing. The depressing effect of aerosols on air temperature is calculated using two methods: first-order approximations of the direct near-surface air temperature

response to each radiative and thermodynamic component (α, εγ and S are included using this method) and multiple noAA simulations. It can be observed that the methods show similar depressing magnitudes in the supplementary figure S6. If the albedo is overestimated because the effects of the snow-albedo feedback can't be captured by taking the mean value, the temperature anomaly could start to deviate and will likely result in a larger value. Consequently, this will have an effect on the mean of the two methods which is represented in figure 8.

Could you elaborate a bit more on the thought of reasoning behind the assumption of employing the mean values of satellite products for these variables (especially concerning the albedo). What are the exact values and sources of these variables which were used for the analysis and do they correspond with previous studies or observations? Perhaps an analysis of the albedo from the downward shortwave radiation products could be included to visualise the temporal variation of the albedo. It's stated in other research that dust and BC can be responsible for a reduction in the albedo besides the snow-albedo feedback. In this research it's shown that dust shows a decreasing trend since 2000 whereas PM2.5, which is related to air pollution, shows an increasing trend. Could there be a possible relationship between PM2.5 and BC which could also contribute to the change in albedo and would you perhaps consider this in the manuscript or future research? Related to the suggestion of the previous major argument, could you include the statistical information regarding the shaded areas in figure 8, S5 and S6.

Thanks. We assumed that the extraterrestrial condition and surface cover type didn't change much especially at 1 lat/lon degree spatial scale and use surface albedo (α), surface emissivity (εs), and extraterrestrial incoming solar radiation (S) climatology to calculate the depressing effect. The corresponding satellite products and time span are listed in Table 2 marked as a depressing effect in usage column.

We use the mean value of satellite products from 2001 – 2015 to calculate the α, εs and S for time span consistency. We calculate the temporal variation of each variable in summer and calculate the standard deviation to prove the reasonability of our assumptions. Based on figure r1, S and εs don't have significant temporal trend (p-value >0.05) and the standard deviation is small. As the comment mentioned above, in recent years the TP undergoes significant climate change while surface cover types and TOA at 1-degree spatial scale didn't change much, so it's reasonable to use variable climatology in the equation.

[Figure]

Figure r1. Temporal variation of (a) TOA DSR and (b) surface broadband emissivity over the TP since 2001.

For the surface albedo, we followed referee #1's suggestion that replaced the single one satellite product by multiple albedo satellite products (GLASS, CLARA, CERES, GlobAlbedo) to calculate the albedo

climatology of the TP. These albedo products cover different satellite observation sources and they have close climatology estimation at mid-latitude as the former study suggested (He *et al.*, 2014). First, we generated the monthly climatological albedo of each satellite product, and we computed all standard deviations of any possible three climatological albedo combinations at each pixel. Then for each pixel, we chose the product combination that has the lowest standard deviation and calculated the mean value to represent the ground truth. We've added more data description and methodology explanation in the manuscript (**Line 165-183**).

Then we did the same temporal analysis for the combined surface albedo data:

[Figure]

Figure r1. Temporal variation of (c) surface albedo over the TP since 2001.

The combined albedo data also kept stable in recent years. We didn't use satellite products to calibrate upward shortwave radiation (USR) for getting surface albedo because there are few USR surface observations to validate the calibration result. Considering that surface albedo didn't change significantly at 1-degree spatial scale, we think it is reasonable to use albedo climatology as input. We've added the figure r1 into the supplementary.

As for the contradiction of 8.08 Wm-2 and SFig. 5, it's the issue of explanation. We've concluded that aerosol radiative forcing negatively affects the surface radiation budget, so the aerosol forcing in SFig 6 (the old version is SFig 5) is actually increasing, which means negative forcing. Thanks!

We discussed that we would consider the aerosol impact on the surface albedo and indirect function for cloud formation over the TP in the future study. This point has been added into the manuscript, Thanks for your suggestions!

**Major argument 3:**

The final aerosol depressing effect on the Tibetan Plateau climate warming is calculated by taking the average over two data sources where one included and the other ignored the heat exchange with the surroundings. The first-order approximation which consists mainly of remote sensing products ignored the heat exchange with the surroundings. The CMIP5 noAA simulations are assumed to be less reliable but did compute the influence of the interaction with other regions. Thus, it is stated that the remote sensing products had more reliable input than the model calculations but this is not supported by numbers/ statistical tests/ previous literature. Furthermore, it seems counterintuitive because the accuracy of the CMIP5 datasets is improved by the NNLS method. Is it a sound methodology to lump these

two sources of data together for the final depressing effect and assume that the exchange is considered to a certain extent? Personally, I'm not convinced by the assumption that the interaction with the surroundings can largely be ignored. In the introduction it is stated that the Tibetan Plateau is a weak heat sink in winter but a strong heat source in summer which is already indicative for differences between the seasons. Also, it's mentioned that the large-scale orography is crucial for water and heat exchange between the Pacific Ocean and Eurasia

This assumption focusses on the exchange of heat with the surroundings but what about other types of exchanges? Aerosols resulting from air pollution in surrounding areas enter the Asian tropopause aerosol layer by deep convection. From here they are consequently transported to other locations. This is an important pathway for anthropogenic aerosols to enter the Tibetan Plateau, which is thought to be the main cause for solar dimming in this region (Lau et al. 2018). Furthermore, the depressing effect calculation is assuming that the change in air temperature is mainly driven by radiative and thermal processes and the change in atmospheric circulation: advection of cold and warm air masses. Again, related to an interaction with the surroundings. Are these interactions included in the results? Could you elaborate a bit more on the points mentioned above in the reply-to-the-reviewer?

I would like to see the supporting material in the manuscript regarding the statement that remote sensing products have a more reliable input than model calculations. A follow-up point of discussion is then related to taking the mean value of these data sources. Figure S5 shows the mean value of the two datasets (with and without interaction). When the two sources of data are separately added to the figure, it enables a visualisation of how they differ/ relate to each other and what their magnitude is in comparison to the mean value. Furthermore, can you justify why the heat exchange is ignored while substantial differences between seasons are found? The final depressing effect propagates in the calculation of the air temperature anomaly which plays a key role in the interpretation and attribution of the solar dimming phenomenon and its effects on surface warming over the Tibetan Plateau.

Thanks for your opinion. TOA DSR is from satellite product that is the only possible observation, thus we consider using it rather than the TOA DSR from CMIP5 into the estimation. CMIP5 didn't release surface emissivity and we used ASTER Surface emissivity product to get broadband emissivity.

As for the surface radiation products, we proved that surface CERES DSR satellite product has significantly high accuracy than individual model simulations and can capture the variation of site observations (See open discussion response to #2, Q4, https://www.atmos-chem-phys-discuss.net/acp-2019-553/acp-2019-553-AC2-supplement.pdf). Besides, based on limited surface albedo observations we proved that the combined surface albedo satellite product has high accuracy than individual model simulations. We've added the figure r2 into the supplementary.

[Figure]

Figure r2. Taylor diagram of solar validation of CERES EBAF (**black dot C**) and 18 CMIP5 models (**grey dots**) based on the CAMP network.

We used calibrated DSR with TOA DSR to estimate the atmospheric shortwave transmissivity and as we mentioned the surface upward radiation is not calibrated due to limited surface validation data, and we also proved the surface albedo didn't change much. Therefore, we use albedo climatology as input and there is no contradiction between high accuracy of calibrated DSR results and low accuracy of low CMIP5 TOA DSR and albedo data.

$$\Delta T_a = 1/f \ (S(1-\alpha)\Delta\tau - S\tau\Delta\alpha - \lambda E + \varepsilon_s\sigma T_a^4\Delta\varepsilon_a$$
$$+ \rho C_d((T_s - T_a)/r_a^2)\Delta r_a) + \ \Delta T_a^{cir},$$

where the *f* is:

$$f = \rho C_d/r_a + 4\varepsilon_s\sigma\varepsilon_a T_a^3,$$

We used the first-order approximation to estimate the depressing effect of aerosol loading and assumed that near the surface, the Ta change is mainly affected by near-surface radiation and thermal process. In the equation, we ignored the influence from surrounding areas because we would like to express that we only focused on aerosol radiative interaction with Ta (the first item, $1/f \ S(1-\alpha)\Delta\tau$) on Ta, $\Delta T_a^{cir}$ and evapotranspiration parts in the equation are ignored. $\Delta T_a^{cir}$ does have a considerable impact on Ta, but for the aerosol radiative process, we consider that the convective transportations of heat and energy have little impacts on aerosol radiative process. Advective transportation can load more aerosols but it was already demonstrated by the variation of atmospheric transmissivity.

For the aerosol radiative effect, aerosols mainly scatter or absorb the direct and diffuse (mainly direct) downward shortwave radiation. It is possible that the surrounding diffuse light can affect the target pixel by scattering more diffuse light, but we considered it ignorable at 1 lat/lon degree. Besides, these two methods didn't have a significant magnitude difference, therefore, we think the assumption is acceptable. We revised the manuscript to clarify that this method mainly focuses on aerosol radiative effect and the

statement "it ignored the heat exchange with surrounding areas" was deleted because heat exchange has little correlation with aerosol radiative process part and will mislead readers.

**MINOR ARGUMENTS**

**Minor issue 1:** There is a difference in the validation of shortwave and longwave radiation due to a system bias at the GAME and CAMP networks caused by disparate instruments. The manuscript states that it's a "minor validation difference" but could you please provide a quantification of the difference?

Thanks! We added the quantification in the manuscript. The minor validation RMSE difference (4.68 Wm-2 in DSR and 9.18 Wm-2 in DLR) between the two networks is the system bias mainly caused by disparate instruments and different site numbers.

**Minor issue 2:** Can the spatial mismatch between radiation datasets and site observations be ignored, even though this is in line with former studies? Especially because the results of this study focus on spatiotemporal variation over the Tibetan Plateau it seems a bit counterintuitive to accept a spatial mismatch in data validation.

Yes, for the downward radiation, we consider the spatial mismatch can be ignored. This is because the downward radiation is hardly affected by the surface heterogeneity. The former study also did the analysis for the downward radiation about spatial mismatch issue (Schwarz *et al.*, 2017), it turns out the sites can represent large spatial areas. In fact, the RMSE results in the study include the uncertainty from the spatial mismatch issue, but comparing with other products that have similar spatial scale, our calibrated datasets have better validation results.

**Minor issue 3:** Firstly, it is stated that deep convective clouds have little influence over the Tibetan Plateau whereas further in the text it is described that aerosols enter the Asian tropopause aerosol layer by deep convection. I assume that deep convection occurs in the regions surrounding the Tibetan Plateau and the aerosols are consequently transported. Thus, deep convective clouds are important but in an indirect pathway.

Thanks. We found that the locations of deep convective clouds are very limited (scattering at some pixels) and it's hard to affect the whole TP. It's a reasonable inference that deep convective clouds have an indirect influence on the dimming over the TP and the advective transportation process has been demonstrated in the former study (Lau *et al.*, 2018). Currently, we focus on the direct effect and don't aim to link all the vertical convective interactions with deep convective clouds. This issue can be discussed in future study.

**Minor issue 4:** The overall variation of multiple models (AA and noAA simulations) is of significance tested in temporal analysis and optimal fingerprinting method. However, no values of a statistical test are given.

Thanks! We added the significance level. The p-value of the impact factor in 1950 – 2005 (1970 - 2005) is 0.216 (0.042).

MINOR ISSUES

**Page 1, line 12**: missing "a" before "higher accuracy"

Thanks! We've revised it.

**Page 1, line 15**: "the fastest decrease in DSR is in the southeastern TP". Maybe it looks nicer to write that the fastest decreases occurs/ can be found in the southeastern TP.

Thanks! We've revised it.

**Page 2, lines 31 and 33**: Firstly, increased surface air temperature is mentioned in line 31. Afterwards in line 33 this suddenly becomes surface temperatures. Is the same variable meant here or are we talking about two different things?

Thanks! We've revised all the related issues.

**Page 2, line 41**: Please be careful with the word "significantly" when it's not supported by a value or reference.

Thanks! We've revised it.

**Page 2, lines 42 and 43**: I would recommend being consistent with terminology. In the abstract and in line 36 the term is introduced as solar dimming whereas in these lines it's mentioned as TP dimming.

Thanks! We've revised them.

**Page 2, line 47**: missing "the" before "TP"

Thanks! We've revised all the related issues.

**Page 2, line 48**: "spatial temporal variation" is used whereas elsewhere in the paper the word spatiotemporal variation is used. Or say: "spatial and temporal variation".

Thanks! We've revised it.

**Page 3, line 72**: It's stated that datasets are chosen which have a spatial resolution less than 2⁰. However, in Table 1 there are two datasets which have a resolution of 2.50⁰ x 1.88⁰ and one with 2.50⁰ x 1.26⁰. Are these datasets not used in the analysis? If they are not used it might be better to remove them from the table.

Thanks! They are used in the analysis because we selected the models that at least one dimension is less than 2 degrees. We've revised the statement.

**Page 3, line 84**: Perhaps it's better to move the link to the reference section of the manuscript. It seems out of place here.

Thanks! We've replaced all URLs to references.

**Page 4, line 103**: Is spatiotemporal resolution meant, or spatial and temporal resolution?

Thanks! We've revised it.

**Page 5, line 130**: I think that "lack" should become lacking. Or "because the sensor calibaration lacks long-term stability".

Thanks! We've revised it.

**Page 5, line 150**: missing "a" before "comparable accuracy"

Thanks! We've revised it.

**Page 6, line 166**: Perhaps it's better to move the link to the reference section of the manuscript. It seems out of place here.

Thanks! We've revised it.

**Page 6, line 168**: Perhaps it's better to move the link to the reference section of the manuscript. It seems out of place here.

Thanks! We've revised it.

**Page 6, line 171**: I would phrase the beginning of this sentence slightly different. Perhaps "collected data from 5 GEBA sites" or "included 5 sites from the GEBA network".

Thanks! We've revised it.

**Page 6, line 173**: I would phrase this sentence slightly different. Perhaps "even though the number of sites is not large enough...".

Thanks! We've revised it.

**Page 7, line 194**: "Given that radiative fluxes are always positive, ....". What kind of sign convention is used here? Usually downward directed fluxes are positive whereas upward direction fluxes are negative (a loss for the surface).

Thanks! We've revised it.

**Page 10, line 278**: compressing does not seem like the right word in this context. Perhaps counteracting or diminishing the greenhouse effect?
Thanks! We've revised it.

**Page 10, line 279**: missing "the" before "TP"
Thanks! We've revised it.

**Page 11, line 319**: missing "a" before **"**different conclusion"
Thanks! We've revised it.

**Page 12, line 339**: missing "the" before "TP" **Page 12, line 340**: missing "the" before "TP"
Thanks! We've revised it.

**Page 12, line 343**: missing "the" before "TP"
Thanks! We've revised it.

**Page 12, line 357**: "causing **a lower** elevation in the model than **in** reality"
Thanks! We've revised it.

**Page 26, Figure 1**: The elevation map which is plotted as background has a scale ranging between 0 and 9000 m. It's difficult to figure out at which location the individual ground networks are located. Could you please add a scale which is better to read?
We redrew the Figure 1, thanks!

**Page 26, Figure 1**: In the central Tibetan Plateau, the network is quite dense and the symbols overlap each other. Could it be possible to provide a zoom-in on this specific area?
We redrew the Figure 1, thanks!

**Page 26, Figure 1**: The caption became very long because all the projects are mentioned by their full name instead of the abbreviation.
We could but the abbr. needs to be explained when the figures are separated from the main text. Therefore, we didn't use the abbreviation.

**Page 28, Figure 3**: The Shi and Liang data covers a relatively small amount of time in comparison with the CMIP5 data. Therefore, I think that adding the regression (for the short period only) is not adding a lot of extra surprising information because the trend is already quite obvious from the time-series. In addition, no statistics regarding the regression are mentioned.
Thanks. We've deleted the regression.

**Page 28, Figure 4**: The caption doesn't mention which data is used for Surface DSR, DLR and Mean Air Temperature. This is mentioned for the air temperature data obtained from ground measurements.
Thanks. We added the information.

**Page 28, Figure 4**: In the caption of the figure suddenly a p-value of <0.01 shows up which is not clearly mentioned in the manuscript.
We've added the significance level in the manuscript. Thanks!

**Page 29, Figure 5**: In all four panels are linear regression lines added, again without any extra statistical information.
We've added the significance test statement in the caption. Thanks!

**Page 29, Figure 6**: The shaded area is not explained in the caption. Are these confidence intervals? For the second panel, the link with the methodology can be a bit stronger so it becomes clear that this figure belongs to the optimal fingerprinting method.
We added more explanations in the caption. Thanks!

**Page 30, Figure 7**: There is a regression line plotted but there are only four points in the figure, and again no statistical significance mentioned.
We deleted the regression lines because we would like to show the clear difference between summer and other seasons in this figure and the regression lines are useless.

**Page 30, Figure 7**: I would have phrased the first line in the caption different because now it seems that the changes are variable instead of variables which are changing.
We revised it.

**Page 7, Figure 7**: In the manuscript, only an explanation is given for the summer season while the other three seasons are plotted as well. Why is it useful to leave them in the figure when nothing is mentioned about them?
Thanks! We would like to show the difference between summer and other seasons. They are left to be compared with summer points. We don't aim to explain all the points and would like to figure out the fastest dimming season and demonstrate the possible reasons.

**Page 7, Figure 8**: The data source is not completely clear to me from the figure caption. Additionally, extra lines which are not represented in the legend are present in the figure (yellow, light-blue and orange). What do these lines represent? The caption and legend should have provided this information. Finally, the shaded area is not explained in the caption. Are these confidence intervals?
The shaded areas in the study are the standard deviation of model average. We've revised the captions and the figure. Data source in Figure 7 is added (CMIP5 model average) and data source in Figure 8 is explained in the corresponding methodology (satellite products used in the first-order approximation method are listed in table 2 and auxiliary meteorological variables like wind, and relative humidity are from CMIP5 historical experiments. NoAA derived air temperature (Ta) data in another method are from CMIP5 HistoricalMisc experiments; and Historical Ta is from the average of four air temperature datasets). Thanks for providing your concern.

**Supplementary material:**

**Page 4, Figure S2 and S3**: The statistical quantification is lacking for the regression (e.g. $R_2$-values).

Thanks. We've revised it.

**Page 5, Figure S5 and S6**: The shaded areas represent uncertainties but it's not mentioned how large these uncertainties are. Is it the 5%-95% confidence interval? The light-red colour in Figure S6 is difficult to see.

We've revised the caption and figure. Thanks.

**References**

He, T., S. Liang, and D. X. Song (2014), Analysis of global land surface albedo climatology and spatial－temporal variation during 1981－2010 from multiple satellite products, Journal of Geophysical Research: Atmospheres, 119(17), 10,281-210,298.

Kuang, X. X., and J. J. Jiao (2016), Review on climate change on the Tibetan Plateau during the last half century, J Geophys Res-Atmos, 121(8), 3979-4007, doi: 10.1002/2015jd024728.

Lau, W. K. M., C. Yuan, and Z. Li (2018), Origin, Maintenance and Variability of the Asian Tropopause Aerosol Layer (ATAL): The Roles of Monsoon Dynamics, Sci Rep, 8(1), 3960, doi: 10.1038/s41598-018-22267-z.

Schwarz, M., D. Folini, M. Z. Hakuba, and M. Wild (2017), Spatial Representativeness of Surface－Measured Variations of Downward Solar Radiation, Journal of Geophysical Research: Atmospheres, 122(24), 13,319-313,337.

Yang, K., B. H. Ding, J. Qin, W. J. Tang, N. Lu, and C. G. Lin (2012), Can aerosol loading explain the solar dimming over the Tibetan Plateau?, Geophysical Research Letters, 39, doi: Artn L2071010.1029/2012gl053733.

---

## Author Comment (AC4) · 11 Nov 2019

Dear Editor and the reader,

We've replied the comments in SC1.

Thank you again for your help!

Best, Aolin Jia and co-authors
* * *